# Long-read powered viral metagenomics in the oligotrophic Sargasso Sea

Joanna Warwick-Dugdale [1,2,9] ✉, Funing Tian [3,9], Michelle L. Michelsen[1], Dylan R. Cronin[3,4], Karen Moore [1], Audrey Farbos[1], Lauren Chittick[3], Ashley Bell [1], Ahmed A. Zayed [3,4], Holger H. Buchholz[1,5], Luis M. Bolanos [1], Rachel J. Parsons[6,7], Michael J. Allen [1], Matthew B. Sullivan [3,4,8] & Ben Temperton [1] ✉

Dominant microorganisms of the Sargasso Sea are key drivers of the global carbon cycle. However, associated viruses that shape microbial community structure and function are not well characterised. Here, we combined short and long read sequencing to survey Sargasso Sea phage communities in virus- and cellular fractions at viral maximum (80 m) and mesopelagic (200 m) depths. We identified 2,301 Sargasso Sea phage populations from 186 genera. Over half of the phage populations identified here lacked representation in global ocean viral metagenomes, whilst 177 of the 186 identified genera lacked representation in genomic databases of phage isolates. Viral fraction and cell-associated viral communities were decoupled, indicating viral turnover occurred across periods longer than the sampling period of three days. Inclusion of long-read data was critical for capturing the breadth of viral diversity. Phage isolates that infect the dominant bacterial taxa *Prochlorococcus* and *Pelagibacter*, usually regarded as cosmopolitan and abundant, were poorly represented.

Bacteriophages are major drivers of both biogeochemical cycles and fitness selection of ecotypes. Ocean viruses directly influence availability of carbon via host lysis[1]; structure microbial communities through negative density dependent selection[2]; and alter biochemical function through co-evolution[3–5] and metabolic hijacking/reprogramming[6,7] (reviewed in[8,9]). Global ocean datasets characterising microbial communities have enabled machine learning and ecosystem modelling approaches to identify which of the many marine microbiota best predict key ecosystem features, including identifying viruses as the best predictor of carbon flux from the surface to deep oceans[10]. In the Sargasso Sea, abundance of virus-like particles has seasonal and depth-associated structure with a maximum concentration observed at 80 m.

Viral abundance correlates positively to abundance of the dominant phototrophs, *Prochlorococcus*, and negatively to abundance of the dominant heterotrophs, SAR11[11]. Curiously, pelagiphages (phages that infect SAR11) have been reported as globally ubiquitous and abundant[12–15], but do not contribute significantly to the variance in virus-associated carbon export to depth in the oligotrophic ocean, which is primarily driven by phages infecting *Synechococcus*[10]. This appears to raise a paradox: Pelagiphages dominate global oceans in abundance, yet appear insignificant in both driving carbon export and restructuring cellular communities in the Sargasso Sea?

One hypothesis of the disconnect between host turnover and viral abundance in SAR11 is that chronic infection is more prevalent than

[1]School of Biosciences, University of Exeter, Exeter, Devon EX4 4SB, UK. [2]Plymouth Marine Laboratory, Plymouth, Devon PL1 3DH, UK. [3]Center of Microbiome Science and Department of Microbiology, Ohio State University, Columbus, OH 43210, USA. [4]EMERGE Biology Integration Institute, Ohio State University, Columbus, OH 43210, USA. [5]Department of Microbiology, Oregon State University, Corvallis, OR 97331, USA. [6]Bermuda Institute of Ocean Sciences, St.George's, GE 01, Bermuda. [7]School of Ocean Futures, Arizona State University, Tempe, AZ, US. [8]Department of Civil, Environmental, and Geodetic Engineering, Ohio State University, Columbus, OH 43210, USA. [9]These authors contributed equally: Joanna Warwick-Dugdale, Funing Tian. ✉e-mail: jo.warwick@gmail.com; b.temperton@exeter.ac.uk

virulent lysis in SAR11 host-virus dynamics, as suggested by low transcriptional activity of pelagiphages in a temperate coastal system[16]. Alternatively, due to the enormity of SAR11 populations, a small proportion of susceptible cells within a much larger population of resistant cells could sustain large pelagiphage populations, as long as susceptible hosts possessed some ecological advantage over resistant conspecifics, as observed in *Synechococcus*[17]. More recently, observations of very low levels of *Prochlorococcus* infection (0.35–1.6%) in oligotrophic waters with high cyanophage abundance has been hypothesised to result from a combination of host resistance, low phage adsorption rates and rapid loss of infectivity of virions[18]. To date, our understanding of the structure of the viral communities in the Sargasso Sea have been limited to seasonal patterns in abundances of viral-like particles[11] and genomic analysis of isolated phages[14,19–22].

Here, we used metagenomics to characterise the viral communities at 80m and 200m in both the cellular and viral fractions of the stratified Sargasso Sea during a four-day cruise in July 2017. Long read viral fraction sequencing was used to overcome assembly fragmentation due to microdiversity and to improve recovery of virally encoded hypervariable regions (HVRs) to facilitate evaluation of their role in niche-adaptation[23–26]. We compared viral abundance between paired cellular and viral fractions to show that the composition of the viral fraction population did not reflect that of the associated cellular fraction. Viral communities were distinct between depths and comprised many viral populations that were undetected in previous global ocean viral surveys. Phages known to infect SAR11 and *Prochlorococcus*, from both isolates and metagenomic viral contigs where host could be determined, were poorly represented in the viral fractions, hinting at potential low viral contribution to cellular turnover and nutrient recycling in these taxa during the sampling campaign.

## Results
### Overview
In marine microbial communities, replication is known to be linked to diel cycles, as observed in both photosynthetic- and heterotrophic bacteria, and picoeukaryotes[27,28]. Assuming that diel cycles would also influence phage production, we sampled cellular and viral fractions for metagenomics over three diel periods to maximise diversity of recovered phage genomes from the Sargasso Sea. DNA was collected from both the cellular fraction ($>0.2 \mu m$; $n = 12$; included host DNA, lysogenic viruses, actively replicating viruses, and any free viruses attached to cells; 0.22 μm filtration and phenol-chloroform extraction method), and the "viral fraction" ($<0.2 \mu m$, $n = 12$; includes virus-like particles; ferric chloride flocculation and spin column extraction method). Samples were taken at depths of 80 m (the 'viral particle maximum'[11]), and 200 m (the mesopelagic), to maximise the diversity of viruses surveyed. DNA recovered from all samples was sequenced to a depth of 12.1 Gbp on the Illumina platform to generate short reads. Additionally, viral-fraction samples from 80 m were sequenced via Nanopore to generate long reads ($n = 3$, 14.7 Gbp total). Short reads were assembled alone and used for hybrid long- and short-read assembly ('VirION'[23]) before identification of viral contigs via VirSorter[29] (all bioinformatic processes are described in detail in Methods and Materials and summarised in Supplementary Fig. 1).

## Results and discussion
### Inclusion of long reads improves virus population recovery
In total, 3514 putative viral contigs >10 kb in length were recovered from the Sargasso Sea; 2049 of these were derived from short-read assemblies (12 cellular fraction and 12 viral fraction samples), and a further 1465 were generated by assembly of VirION reads (three viral fraction samples). Contigs were clustered into 2301 viral populations[24] (equivalent to species). In 1410 (61%) of the viral populations the longest contig (selected as the cluster representative) was derived from long-read sequencing of pooled 80 m samples. Only 163 (7%) of

the viral populations contained both long- and short-read derived contigs. No population clusters with two or more members were comprised exclusively of long-read contigs, suggesting either the coverage of the virome in long read data was low, or that short-read sequencing captured genomes across all viral populations, but such genomes were fragmented and long-read assembly provided longer representatives. When the 24 short-read assemblies were processed without long reads, 1044 viral populations were identified, indicating that 55% (1257) of the total viral populations reported here were only captured with the inclusion of long-read sequencing.

An important question is whether viral populations represented by long-read assembled contigs were successfully recovering contigs from short-read assemblies that had fragmented below the 10kb cutoff, or whether the additional sequencing depth offered by long-reads enabled better recovery of 'rare' viruses. If the former is true, fragmented contigs from abundant viruses should align to the genomes of their respective long-read representative. If the latter is true, clusters with long-read representatives should be enriched at low relative abundance values – for viral populations with high relative abundance, there would be sufficient data to recover the genome without the addition of long reads. Mapping of short-read data to viral population representatives indicated viral populations with long-read representatives were abundant in the Sargasso Sea (Fig. 1A). Alignment of short-read population cluster members to long-read population cluster representatives illustrated fragmentation and/or poor overall recovery of 115 viral population genomes across a range of relative abundances in the short-read assemblies (Supplementary Fig. 2A). If assembly of viral genomes from short read data is restricted by genome coverage, one would expect that the sum of fragment lengths from abundant viruses would be similar to that of the long-read representative of their cluster, whereas low abundance viruses would cover less of the long-read representative genome. However, the extent to which genomes from long read assemblies were recovered by mapped short-read viral contigs did not correlate to the relative abundance of those contigs, when either viral contigs >10kb were used (Supplementary Fig. 2B), or when short-read contigs >1kb were used (Supplementary Fig. 2C) suggesting assembly breakages were not a result of low coverage. Representative contigs from long-read assemblies were 38.5% longer than those from short-read assemblies (at median lengths of 17,372 bp and 15,755 bp, respectively; two-sided Mann-Whitney U test, $p < 0.001$) (Fig. 1B). Inclusion of long-read data was also critical for enabling the recovery of hypervariable regions in viral genomes predicted to encode proteins involved in host recognition, DNA synthesis and DNA packaging (Supplementary Table 1; Supplementary Disscussion). Together, these results suggest that long-read sequencing of viromes enhanced the capture of both different and more complete viral genomes from the Sargasso Sea compared to short-read technology alone.

### Sargasso Sea Viruses formed a distinct community
We postulated whether Sargasso Sea viruses were endemic or globally distributed. We calculated the distribution of Sargasso Sea viruses from global oceanic viral metagenomes (GOV 2.0[30]), and found that 800 (45.7%) of the viral genomes from this study were represented within other subsampled (to 5 million reads) global oceanic viral metagenomes (GOV 2.0) (Fig. 2). The log odds of finding a virus from the Sargasso Sea represented in a GOV 2.0 sample from a temperate/tropical epipelagic sample was negatively correlated to the availability of nitrate/nitrite and phosphate (logistic regression, nitrite/nitrate log odds = −0.13208; phosphate log odds = −0.13208, $p << 0.05$ for both), suggesting greater frequency in samples from similarly warm oligotrophic oceanic regions (e.g., TARA_R100000455; Fig. 2). Sargasso Sea viruses were absent from polar regions, consistent with previously reported ecological patterns of ocean viral communities[30]. However, 951 (54.3%) of the Sargasso Sea viral populations were not represented

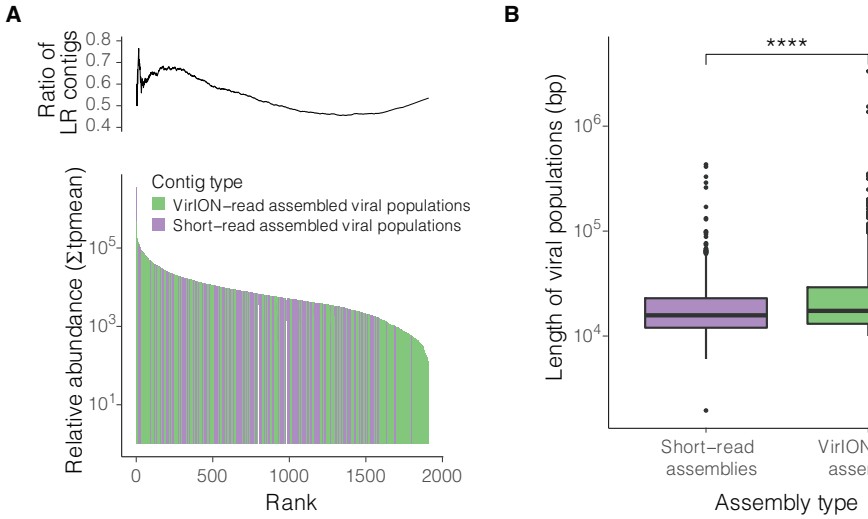

**Fig. 1 | BATS viral populations. A** Histogram of sequencing-depth adjusted coverage for viral populations ($n = 2301$). Long-read sequencing was able to rescue more viral populations, and these viral populations were abundant ($\sum$ tpmean: sum of mean relative abundance of viral populations excluding < 5th and > 95th percentile). Inset: Cumulative ratio of viral populations derived from long reads (LR). **B** Boxplots showing statistically significant differences in lengths of viral representative contigs assembled from short reads ($n = 891$) and VirION reads ($n = 1410$) (median lengths 15,755 bp and 17,372 bp, respectively; two-sided Mann-Whitney U test, ****$p = 8.08 \times 10^{-9}$). Boxes represent upper and lower quartiles; whiskers represent 1.5x interquartile range; individual points represent outliers; Centre line represents median population length.

in the global ocean dataset, suggesting that these viruses are either endemic to this subtropical region, or are enriched at this site and below the detection limit elsewhere. Twenty-four of these 'endemic/enriched' viruses were among the top 100 most abundant viruses at this site, indicating their prevalence in the Sargasso Sea. In contrast, our previous application of long-read sequencing to coastal Western English Channel viruses revealed that viral contigs obtained from a single sample were abundant in global marine viromes[23].

Inter-community diversity of the Sargasso Sea viral populations was compared to previously established patterns in GOV2.0[30], and revealed that Sargasso Sea viral communities have a distinct community structure (analysis of variance: Adonis F-test; *p*-value = 0.001; Fig. 3). Bathypelagic GOV2.0[30] samples were very divergent (Supplementary Fig. 3A, B), supporting previous evidence that viral populations from deep samples are distinct from others[30]. To improve resolution bathypelagic samples were removed (Fig. 3), as were three GOV 2.0 viromes (station 155_SUR, station 72_MES, station 102_MES) that were outliers in both PCoA and Shannon' H analyses[30]; Supplementary Fig. 3B). Sargasso Sea viral populations (sampling depth: 80 m and 200 m) were most similar to viromes from other temperate-tropical mesopelagic regions (Fig. 3; two-sided pairwise Adonis F-tests of Euclidean distance between centroids (viral fraction populations vs TT-MES: 0.155; cellular fraction populations vs TT-MES: 0.101); *p*-value = 0.001; Supplementary Dataset 1). A Similarity Percentages (SIMPER) analysis was performed to determine the key Sargasso Sea viruses contributing to the dissimilarity of this group from viruses from the other ecological zones sampled in the Global Ocean Virome (GOV2) dataset (Fig. 4; Supplementary Dataset 2). SIMPER analysis showed that 754 viruses captured in the long-read data explained 9.5% of the variance that discriminated Sargasso Sea viromes from the other viral communities/zones. Viral populations were classified into two sets (Set A and Set B) depending on their recruitment of viral reads from other GOV2 samples. A set of 80 viral populations from the Sargasso Sea (Set A) recruited reads from other global viromes and were important in discriminating between temperate and tropical epipelagic (TT-EPI), and temperate and tropical mesopelagic (TT-MES) populations, and between TT-EPI and arctic (ARC) populations. Within these viral populations, 79 out of 80 were comprised of singleton viral populations from long-read assembly that lacked even fragmented

short-read assemblies. The median number of Global Ocean Virome (GOV2) samples in which a phage in Set A was observed was 56 (52.5–58 95% CI; out of 142 total). Therefore, we hypothesise that these viruses are common across oceanic regions but missed from existing short-read viral metagenomic datasets. Five of these viral populations had greater global relative abundance than pelagiphage HTVC010P and 51 were more ubiquitous than HTVC010P and other isolates infecting SAR11 and *Prochlorococcus* spp., at middle and lower latitudes (Fig. 4). These observations suggest that the VirION approach captured globally distributed and ubiquitous viruses that would otherwise have remained unidentified. The remaining 675 viruses (Set B), were far less ubiquitous in global oceans, identified in a median of 10 (4.5–17 95% CI) of 145 GOV2 samples. Four members from Set B recruited a large number of reads from at least one site in either the TT-MES or Antarctic biomes (Supplementary Dataset 2), implying some degree of viral import from either upwelling or ocean currents into the 80m and 200m samples. Overall, these results show that the viral community of the Sargasso Sea was distinct in the global ocean, for at least the duration of the 3-day sampling campaign, and supports the idea that ubiquitous viruses may contribute to the regionalisation of viral communities through their relative contribution to overall community structure[31].

## Viral communities were decoupled by fraction

The physical stratification of the ocean due to seasonal warming (and therefore reduced density) of the upper layers, combined with the attenuation of light with depth, has long been understood as an important factor in structuring pelagic, microbial communities down the water column[32–35]. Likewise, the viral community composition in the Sargasso Sea differed significantly along the two ecological zones, as delineated by depth (Fig. 5A). These results support previous evidence that phages are vertically distributed[30] in the same way as their bacterial counterparts, corresponding to the stratification of the water column during summer[36]. Further investigation comparing these results to viral communities sampled after the spring bloom would ascertain whether the same pattern is repeated throughout the year, when vertical stratification is either absent or minimised.

Here, we report significant differences in composition and membership of viruses between cellular and viral fractions from the same

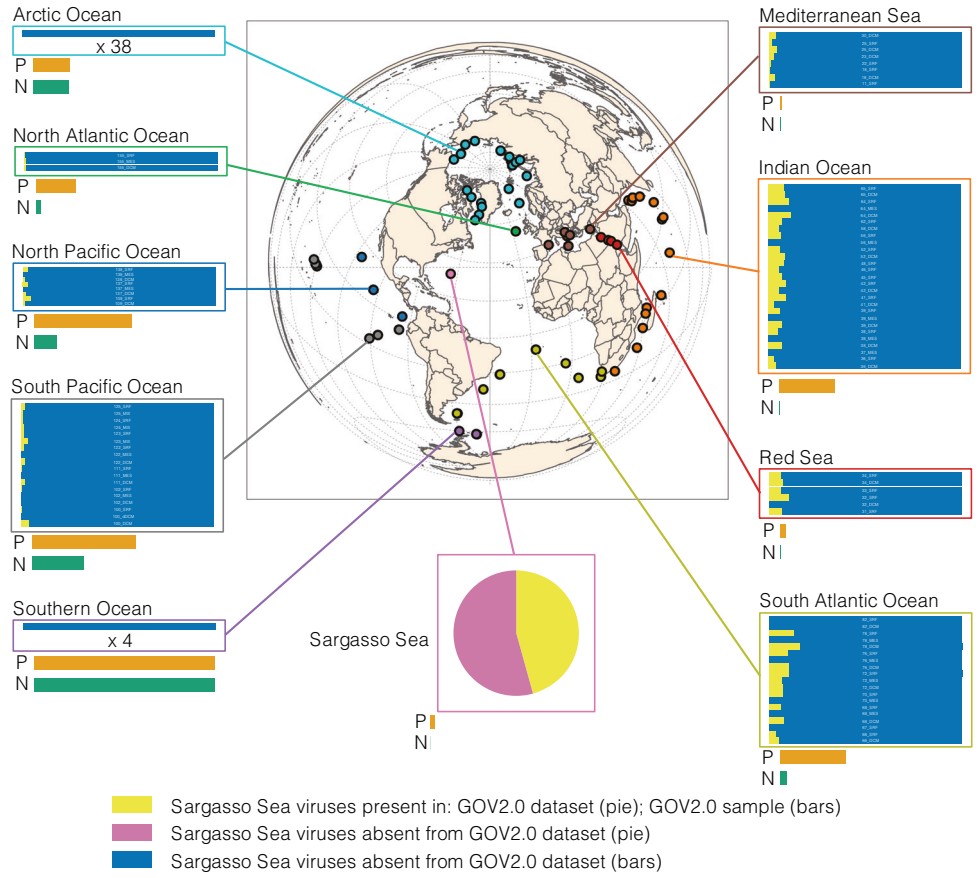

**Fig. 2 | Global distribution of BATS viral populations.** Dereplication of the combined Global Ocean Virome 2[30] (GOV 2.0) and Sargasso Sea dataset resulted in 152,979 viral populations (circular contigs or those with lengths ≥ 10 kb). Presence/absence of Sargasso Sea viruses are shown in various oceanic regions (within associated boxes). Under half of Sargasso Sea viruses (45.7%; coloured yellow at all sites) were observed as present elsewhere in the world's oceans. These viral sequences were identified as present in the GOV 2.0 dataset via competitive read recruitment of subsampled (5 million per sample) reads (mapped at ≥ 90% read length at ≥ 95% identity) with > 70% genome coverage. The majority of Sargasso Sea viruses were not observed at GOV 2.0 sites (54.3%; coloured pink at the Sargasso Sea site; coloured blue at other sites), and appear endemic to the Sargasso Sea, (i.e., below the level of detection in the subsampled GOV 2.0 dataset). Levels of nitrogen (NO$_3$ = Nitrate+Nitrite$^{-1}$ (umol/kg); N; coloured green) and Phosphate (PO$_4$ = Phosphate$^{-1}$ (umol/kg); P; coloured orange) were normalised between sample sites (e.g., lowest site median: P = 0; highest site median: P = 1) and show that more Sargasso Sea virus tend to be present at oligotrophic GOV 2.0 sites.

depth. Bacteriophage populations in the Sargasso Sea formed discrete cellular fraction and viral fraction populations, at both the viral maximum depth (i.e., 80 m[11]) and in the mesopelagic (Fig. 5A). This zonal partitioning was also observed when long-read derived contigs were excluded from the analysis (Supplementary Fig. 4), indicating that it is not an artefact of sequencing technology. Sixty-five percent of viral populations from the short reads (678) were only detected in the 'viral fraction', whereas 7% (74) were discovered solely in the cell-associated fraction, and 28% (292) were detected in both fractions (Fig. 5B). Previous investigations of soil microbial communities and their phages have examined coupled cellular and viral fractions of samples for viral genomes and found that the majority (77%) were present in the cellular fraction, despite the samples for each fraction being collected years apart[37,38]. However, when viromes and cellular fractions of the same samples were compared, as undertaken here, just 9% of viral populations were shared between fractions, with > 90% present in only the viral fraction, and < 1% unique to the cellular fraction[39]. Because the Sargasso Sea cellular fraction here may include any free viruses attached to- or caught on cells during the filtration step of our protocol, it is not possible to discount the presence of DNA from free viruses here. The large surface area of filters used to separate cellular and viral fractions, and the relatively low concentration of bacterioplankton cells during the sampling campaign (80 m: -5–7.7 x 10$^8$ L$^{-1}$; 200m: -1.2–3.2 x 10$^8$ L$^{-1}$) make it unlikely that viruses were removed

from the 'viral' fraction by clogging on filters. Thus, the effect of filtration is unlikely to cause the degree of dissimilarity between the free-particle and cell-associated fractions observed in our data. Presence-absence analyses showed that this decoupling of fractions was less pronounced in the 100 most abundant Sargasso Sea viruses (Supplementary Fig. 5), thus rare viruses may be partially driving the dissimilarity in viral- and cellular fraction viral populations. However, Bray-Curtis analysis is robust to the effects of sample size[40], so the partitioning effect illustrated via Principal Coordinates Analysis is not likely due to under-sampling of rare viruses.

One explanation for the decoupling of cell-associated and viral fraction populations is that the viral fraction population represents the integral of infections in the cellular fraction over time, whereas the cell-associated populations are a mix of prophages, remnant phages, active lytic infections and a small number of phage particles trapped on filters. Viral turnover in oligotrophic waters has been estimated at 2.2 days, compared to 0.82–1.3 days in coastal waters[41]. Here, sampling was conducted over three days to capture viruses across full host growth and lytic cycles to avoid biases inherent to snap-shot surveys, where asynchrony in infection cycles can cause apparent dissimilarities between free and cell-associated viral populations. Decoupling between free and cell-associated viral populations suggests that, at the time of sampling, viral turnover in the Sargasso Sea occurred over periods longer than our sampling campaign. Loss of infectivity and

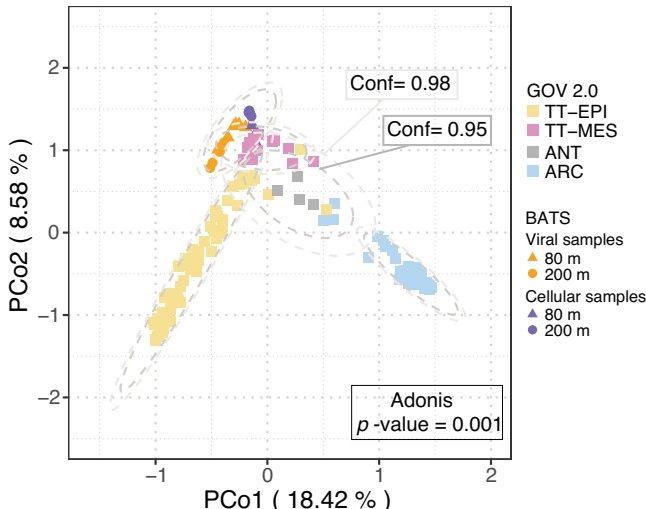

**Fig. 3 | Inter-community diversity of Sargasso Sea and Global Ocean Virome 2 (GOV 2.0) viral populations.** Principal coordinates analysis (PCoA) of a Bray-Curtis dissimilarity matrix calculated from mapping Global Ocean Virome 2[30] (GOV 2.0) reads and Sargasso Sea short reads to a combined dataset of Sargasso Sea and GOV 2.0 viral populations (contig lengths ≥ 10 kbp; n = 152,979); Viral community structure was suggested by ellipses drawn at 95% (inner) and 98% (outer) confidence intervals and analysis of variance (two-sided Adonis, p-value = 0.001); ARC Arctic, ANT Antarctic, TT-EPI Temperate and tropical epipelagic, TT-MES Temperate and tropical mesopelagic (divergent bathypelagic samples removed).

decay of viral particles by sunlight[42] could also contribute to decoupling. However, at the depths sampled here (80 m; 200 m), sunlight-induced viral decay is likely to be lessened compared to surface samples (5 m) where previous turnover estimates have been calculated[41].

A potential contributing factor for decoupling may be that the rate of active lytic viral replication in dominant members of the community such as SAR11 is low in the Sargasso Sea, as was previously observed in a coastal system[16]. This parallels a recent report that high abundances of free cyanophages coincided with low levels of *Prochlorococcus* infection in oligotrophic waters, proposed to result from a combination of loss of infectivity and low adsorption efficiency, alongside host resistance[18]. Success rates of SAR11 cultivation from dilution-to-extinction culturing experiments have been speculated to indicate that a large proportion of streamlined heterotrophs such as SAR11 and OM43 are possibly dormant, and therefore not contributing to viral turnover[43]. In this scenario, viruses are released into the viral fraction but are unable to efficiently re-infect the cell-associated fraction, decoupling the two populations. High rates of lysogeny in taxa such as SAR11 would produce a similar outcome. Phylogenomic and genetic analysis of the abundant HTVC010P-type SAR11 pelagiphages has revealed that many encode an integrase gene (e.g., the isolate HTVC010P-type SAR11 prophage PNP1[44,45]). However, the most abundant pelagiphages in the oceans do not encode known mechanisms of chromosomal integration[14,15,45]. There was no evidence in environmental data collected on the cruise (C. Carlson, pers. comm.) to suggest entrainment of water from outside the gyre, nor do the high numbers of site-specific viral populations in the Sargasso Sea (Fig. 2) support possible decoupling by significant entrainment of viruses from outside the sample site concurrent to the sampling campaign.

### Host prediction remains challenging despite MAGs
To determine viral taxonomic classification, we clustered the 2301 Sargasso Sea viral population representatives for genera-level classification based on shared-gene networks[46]. RefSeq prokaryotic viral genomes (NCBI RefSeq v88 release) were included to assign family-level taxonomy to clusters. Out of 548 viral clusters, 186 contained

Sargasso Sea viral sequences, of which 177 lacked a representative sequence in the RefSeq database, consistent with previous studies of environmental viral communities[31,47–50]. All phages assigned a taxonomy clustered with viruses within the order *Caudovirales* except one genome from family *Microviridae* with the order level *Petitvirales*. Few viral family-level taxonomic annotations were predicted: members of class Caudoviricetes (formerly known as 26 *Podoviridae*; 3 *Myoviridae*; 3 *Siphoviridae*); 1 *Microviridae*. Among the top 50 most abundant viral populations, 8% were classified, all of which were resolved as either *Podoviridae* or *Myoviridae*, highlighting the need for improved representation of abundant environmental viruses within reference databases[50]. We next tried to assign putative hosts to Sargasso Sea viral populations by recovering metagenome-assembled genomes (MAGs) from cellular metagenomes. Assembled contigs were binned using sequence composition, relative abundance, and taxonomical classifications to group contigs into MAGs[51]. In total, we obtained 89 MAGs with ≥ 70% completion and ≤10% contamination that were used for host-prediction (Supplementary Dataset 3). Considering the intrinsic challenges of in silico host prediction[52,53] a scoring matrix was developed to combine the results from prophage blast, tRNA scan, and WIsH to improve the accuracy of host assignments. Despite this, only six out of 2301 viral populations (0.26%) were successfully linked to three hosts which belonged to the phylum of Actinobacteriota (n = 2) and Chloroflexota (n = 1). Therefore, establishing virus-host linkages from metagenomes remains a major barrier for understanding environmental viral ecology.

### Low abundance of known cyanophages and pelagiphages
During the summer months, cyanobacteria (*Prochlorococcus*) are the dominant phototrophs in the Sargasso Sea, with estimated relative abundances of up to 35% in the euphotic zone[34]. Also highly abundant, SAR11 comprises 20–40% of the total bacterioplankton community in open ocean systems[54,55]. Here, amplicon sequencing and analysis revealed high relative abundance of SAR11 at the sampling site over the course of the campaign. SAR11 16S rRNA Amplicon Sequence Variants (ASVs) contributed a maximum 52.7% and minimum 47% of the total amplicons from 80 m, and a maximum 45.6% and minimum 27.7% from 200 m (Supplementary Fig. 6). *Prochlorococcus* ASVs comprised a maximum 11.3% and minimum 4.1% of total amplicons from 80 m, and were not observed at 200 m (Supplementary Fig. 6). Previously, total viral abundance was shown to be negatively correlated with SAR11 abundance and positively correlated with *Prochlorococcus* over seasonal scales, leading to the hypothesis that viral communities were dominated by cyanophages with pelagiphages poorly represented[11]. Composition analysis of the viral fraction here supports a dearth of pelagiphages throughout the sampling campaign. Recruitment of reads to *Pelagibacter* phage HTVC010P (a virus previously cited as the most abundant on Earth and isolated from the Sargasso Sea[14], failed to meet the minimum genome coverage to be classified as present, even at a relaxed cut-off of 40% (Supplementary Fig. 7A, B). Moreover, *TerL* genes identified in the 100 most abundant viral populations did not cluster with known pelagiphage terminase genes within a phylogenetic tree (Supplementary Fig. 8). Thus, we found no evidence in either assemblies or short-read data of abundant viruses in the Sargasso Sea that are closely related to previously isolated pelagiphages. In comparison, the same read recruitment strategy revealed that known pelagiphages were well represented in global epipelagic viromes (GOV 2.0; TT-EPI), although less evident in mesopelagic samples (GOV 2.0; TT_MES) (Supplementary Fig. 7A, B). Conversely, no known pelagiphages were detected at ≥40% minimum genome coverage in any cellular fraction metagenomes from the Sargasso Sea, either from this study or previously published cellular-fraction metagenomes[56] (Supplementary Fig. 7C, D).

Surprisingly, cyanophages were also rare in our Sargasso Sea samples. Among the top 100 most abundant viral populations, only four

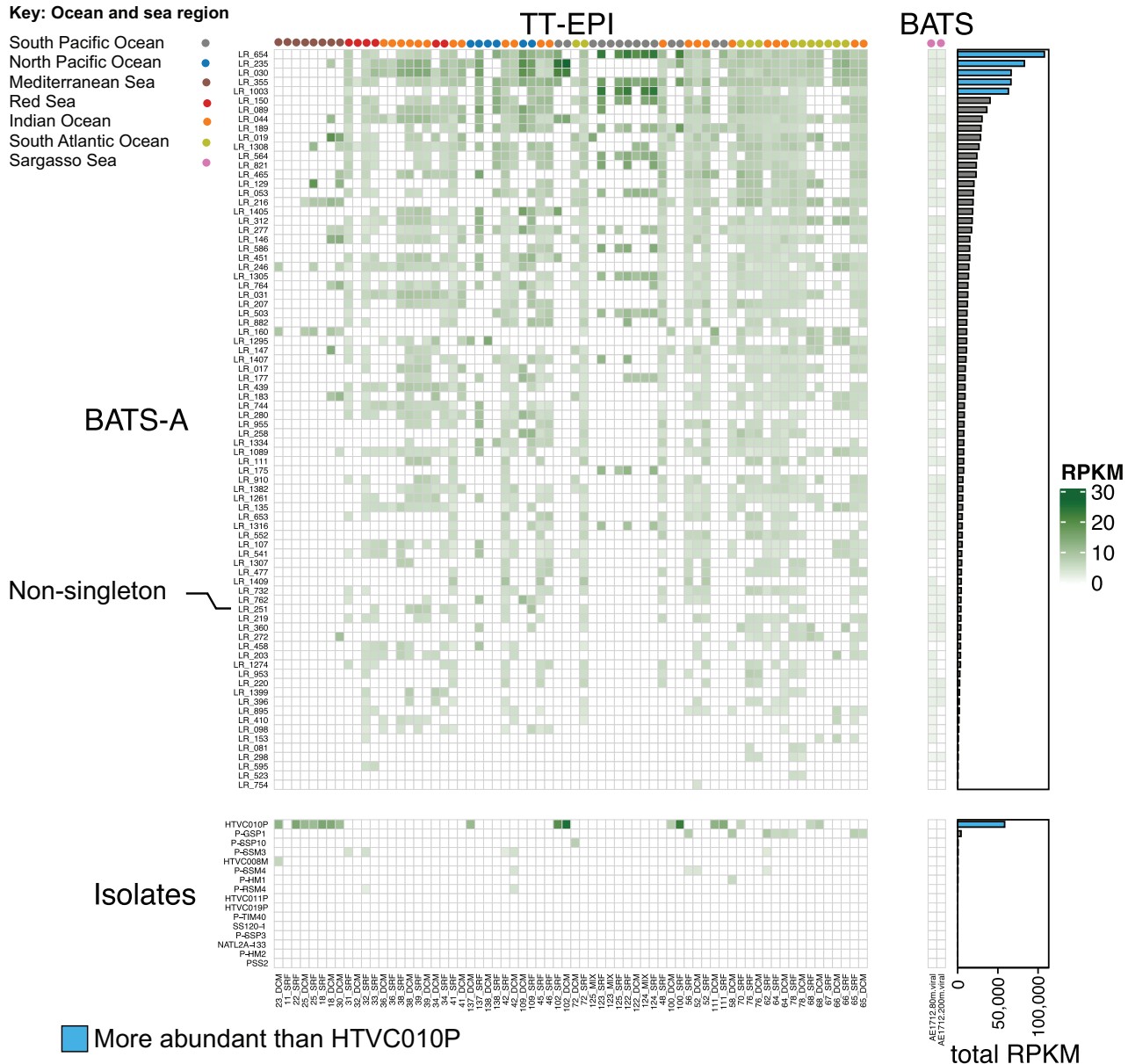

**Fig. 4 | Global relative abundance of Sargasso Sea long-read viral contigs from this study (BATS-A) and isolates that infect SAR11 and *Prochlorococcus*.** BATS-A viruses were important for discrimination between temperate and tropical epipelagic (TT-EPI) and temperate-tropical mesopelagic (TT-MES) viral populations in SIMPER analysis. 79 out of 80 BATS-A viruses were recovered from singleton populations (i.e., they were not captured by short-read assemblies); five of these viral populations had greater global relative abundance (total RPKM) than Pelagiphage HTVC010P in the epipelagic (at middle and lower latitudes). The bootstrapped median ($n = 10,000$) number of Global Ocean Virome[30] (GOV2) samples in which a BATS-A viruses were observed was 56 (52.5–58 95% CI). Thus these viruses are common across oceanic regions, but were missed from existing short-read viral metagenomic datasets.

contigs were identified as potential cyanophages via presence of the *psbA* gene, with just two of these having completeness over 80%; these two populations were the 26th and 45th most abundant in the Sargasso Sea. In addition, when short read sequences from both the viral fraction and cellular fractions were recruited directly to a database of all published cyanophage and pelagiphage genomes, only two isolate genomes (viral fraction: *Synechococcus* phage Bellamy, MF351863.1; cellular fraction: *Prochlorococcus* phage P-SSM2, GU071092.1) were identified as present at a minimum genome coverage of 70% (Supplementary Fig. 7D). Although amplicon data showed that *Prochlorococcus* abundance did not approach the maximums recorded at this site (ref. 34, Supplementary Fig. 6), fewer cyanophages were observed than might be expected if active infection was prevalent during the sampling campaign. Low cyanophage abundance despite a comparatively higher

proportion of potential hosts contrasts with predictions by Parsons et al. [11]. However, low cyanophage abundance may help to explain why Sargasso Sea viral populations from 80m and 200m were more similar to TT-MES samples (depth: 150–1000 m) in the GOV2 dataset (Fig. 3), where cyanophages are also rare, than TT-EPI samples (depth: 0–150 m), where cyanophages are abundant (Supplementary Fig. 7B). It is possible that known cyanophages or pelagiphages were not observed due to limitations in the sampling regime of this study (two depths, in one site over a consecutive four-day period) and that they may be detected at different times/depths/locations within the Sargasso Sea. Cyanophages were better represented in previously published cellular-fraction metagenomes from the Sargasso Sea[56] than in our samples (Supplementary Fig. 7C, D), suggesting the importance of sampling timing and duration within seasonal cycles.

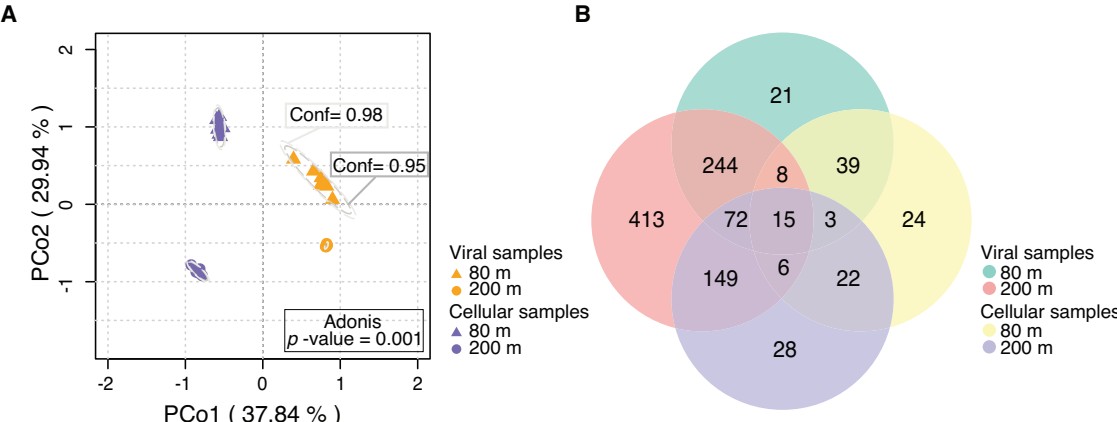

**Fig. 5 | BATS viral populations clustered on the basis of sample fraction type (c = cellular; v = free particulate) and depth (80 m; 200 m). A** Principal coordinates analysis (PCoA) of a Bray-Curtis dissimilarity matrix calculated from reads mapped to BATS viral populations ($n$ = 2301) derived from short-read and VirION assemblies. Viral community structure was suggested by ellipses drawn at 95% (inner) and 98% (outer) confidence intervals and analysis of variance (two-sided Adonis, $p$-value = 0.001). **B** Number of viral populations ($n$ = 1044) found in 80m viral samples, 80m cellular samples, 200 m viral samples, and 200 m cellular samples, as identified from short-read only assemblies.

Rates of successful infection of *Prochlorococcus* in the oligotrophic surface waters of the North Pacific Subtropical Gyre (NPSG) were previously found to be low, despite a high abundance of cyanophage virions ( ~ 2.2% of the viral fraction)[18]. Therefore, it appears that in the NPSG, infection of a small proportion of the cyanobacterial community is sufficient to maintain abundant viral particles. The lack of representation of abundant pelagiphages and cyanophages observed in this study could suggest similarly low levels of active infection at this site in the Sargasso Sea over the duration of the sampling regime. Metabolic processes associated with pelagiphage infection were poorly represented in transcriptomic data from a coastal system collected over a 2-year period ($n$ = 8), suggesting that chronic infection was more prevalent than lytic infection[16]. This, coupled with predicted dormancy rates of up to 85% in SAR11 cultured isolates[43] may further limit efficiency of pelagiphage infection in the Sargasso Sea. Yet in contrast to the NPSG study, we found little evidence of abundant cyanophages in the viral fraction from the Sargasso Sea at the time of sampling. An important difference between these regions is that phosphate concentrations in the Sargasso Sea are up to two orders of magnitude lower than those found in the NPSG[57]. Synthesis of viral particles imposes a high phosphate requirement on infected cells and phosphate limitation restricts lytic infection in both cultures of cyanobacteria and natural marine communities (reviewed in ref. [58]). We propose that during the sampling campaign, the increased P-limitation of the Sargasso Sea reduced viral production to a level where known pelagiphages and cyanophages (either isolates or closely related phage) were diluted to below levels of detection with metagenomic approaches. Here, clustering of the *TerL* genes identified in the 100 most abundant viral populations (Supplementary Fig. 8) implies that the most abundant Sargasso Sea viruses captured are most like those that infect copiotrophic hosts (e.g., *Pseudoalteromonas* and *Flavobacterium* phages). This could possibly suggest that boom-bust, particle associated interactions may be of greater importance than previously imagined in nutrient-limited water. However, as a result of the challenges of host-prediction, highlighted by our results here, this hypothesis is highly speculative at present, and more robust methods of host prediction are required for further investigation.

## Sargasso Sea viral populations are microdiverse
Given that the Sargasso Sea bacterial community has more fine-scale, intraspecific diversity than variation at the species (or macrodiverse) level[34,59], we determined whether the same could be true of their phages, and how levels of microdiversity might compare to those of global oceanic datasets. Here we report high microdiversity (i.e. intra-population diversity; $\pi$[60]) values for Sargasso Sea viral populations across both depths (mean $\pi$: $3.411 \times 10^{-4}$ ($2.473 \times 10^{-4} - 4.334 \times 10^{-4}$, 95% CI) (Supplementary Fig. 9), comparable to those recorded at similar latitudes in the Global Ocean Virome (GOV 2.0[30]). The microdiversity of Sargasso Sea viral populations from 80 m and 200 m samples was not observed as significantly different (permutation and bootstrapping test: $p$ = 0.164; Fig. 6A; Supplementary Fig. 10). This result does not align with those of Gregory et al.[30], who reported higher levels of microdiversity in mesopelagic temperate and tropical viromes than those from the epipelagic in the GOV 2.0 dataset. However, it is possible that the mesopelagic nature of the viral communities captured here work to reduce the signal of increasing microdiversity at depth observed in tropical and temperate viromes[30].

Additionally, we compared microdiversity of Sargasso Sea viral populations obtained using only short reads, and those captured with the inclusion of VirION reads, as previously we have observed that long reads assemble microdiverse viral genomes of Western English Channel viral assemblies better than short-reads due to the benefits of long-read assembly[23]. The average microdiversity values for viruses derived from the two assembly types were $4.213 \times 10^{-4}$ ($3.283 \times 10^{-4} - 5.326 \times 10^{4}$, 95% CI) and $2.03 \times 10^{-3}$ ($1.69 \times 10^{-3} - 2.36 \times 10^{-3}$, 95%CI), for short-reads and VirION reads, respectively (Fig. 6B), which represents an average increase of 389% (264.559 – 551.95%, 95% CI) in the $\pi$ value calculated for viral genomes captured by VirION compared to those assembled from short-reads (permutation and bootstrapping significance test: $p$ < 0.001) (Supplementary Fig. 11). Because $\pi$ values are calculated from short reads which map to viral contigs, rather than the viral contigs themselves, this finding is not influenced by residual error in long-read derived contigs. This result confirms that VirION sequencing facilitates the capture of more viral microdiversity than is possible from short-read sequencing alone.

However, the question as to why oligotrophic regions such as the Sargasso Sea produce highly microdiverse viral assemblages remains open. The low infection rates observed in the NPSG[18] suggests that phages here are not likely the main control on host abundance. Instead, such viruses may play an important role in the evolution of clonal, fine-scale diversity in the host population (as proposed decades ago[17]). Phages have been shown to increase host diversity on this micro sale in host-phage model systems[61], and the high divergence in potential phage recognition sites observed in the HVRs of Pelagibacter

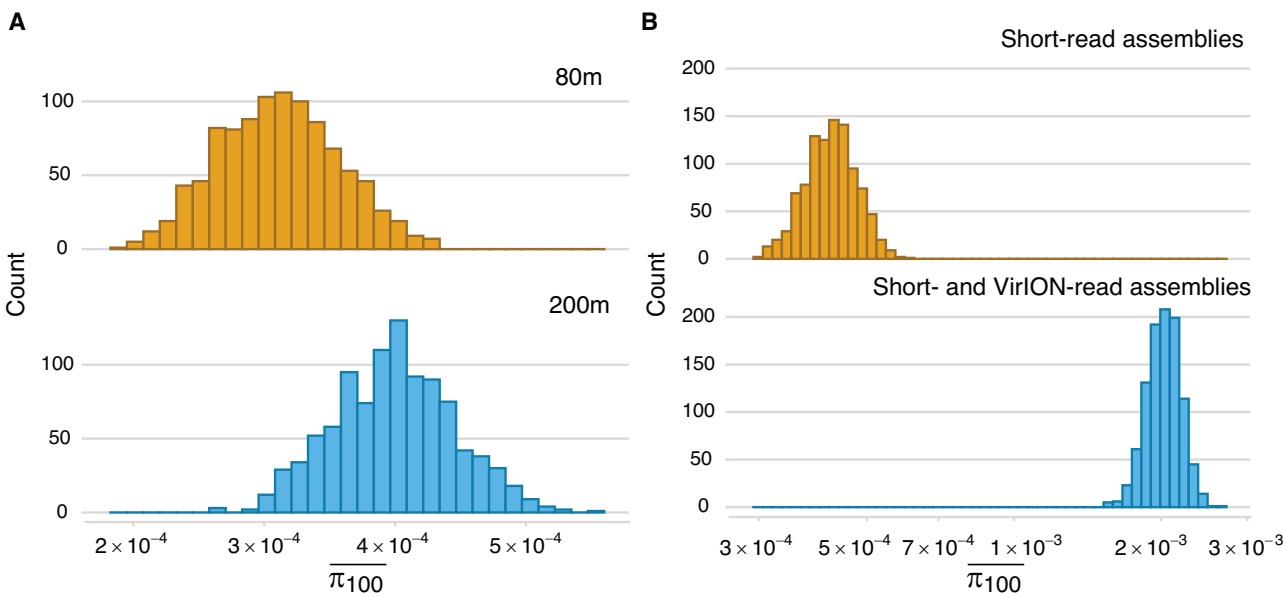

**Fig. 6 | Microdiversity (intra-population diversity; mean π) of Sargasso Sea viral populations.** (Calculated as Gregory et al. [30]): **A** The microdiversity of Sargasso Sea viral populations from 200 m and 80 m samples (mean π: $3.119 \times 10^{-4}$ ($2.347 \times 10^{-4}$ – $4.094 \times 10^{-4}$, 95% CI) and $3.967 \times 10^{-4}$ ($3.072 \times 10^{-4}$ – $4.909 \times 10^{-4}$, 95% CI), respectively). On average, microdiversity was ~30% greater in viral genomes from 200 m than those from 80 m samples, but this was not a significant difference

(permuted significance test: $p = 0.164$; Supplementary Fig. 10B). **B** VirION (long-read assembly and hybrid long- and short-read assembly) facilitated the capture of viral genomes with high microdiversity: mean π was 388.668% (264.559 – 551.95%, 95% CI) greater for viral genomes captured by VirION ($n = 1465$) compared to those assembled from short-reads ($n = 2049$) (permuted significance: $p < 0.001$; Supplementary Fig. 11B).

isolates[2] supports the idea that phages drive host microdiversity in the Sargasso Sea. Because increased host microdiversity will expand phage niches, phage adaptation to emerging host ecotypes may increase phage microdiversity, as predicted by the Diversity Begets Diversity (DBD) theory of species interactions[62,63]. DBD, which posits that existing diversity will promote the evolution of further diversity by niche construction (plus other types of interaction), appears to be most prevalent in low-diversity biomes[63].

## Methods

### Sample collection and DNA extraction

Metagenomic samples were collected aboard the *RV Atlantic Explorer* at the Bermuda Atlantic Time Series (BATS; http://bats.bios.edu/) station (31°40′N, 64°10′W) via rosette-mounted Niskin bottles during dusk (~19:00 local time) and dawn (~06:00 local time), from depths of 80 m and 200 m, over a period of four consecutive days from the 8th–11th July 2017. Host communities ($n = 12$) were obtained from 5 L of seawater per sample transferred immediately to clean polycarbonate bottles; the cellular fraction was recovered onto 0.22 μm pore Sterivex filters via positive pressure filtration. Each sample was stored in the dark at −20 °C in 1 mL of SET buffer (0.75 M sucrose; 40 mM EDTA; 50 mM Tris-base). Within a fortnight of collection, DNA from the cellular fraction (which included host DNA, lysogenic viruses and any free viruses attached to cells) was extracted using phenol-chloroform[64,65], resuspended in 10 mM Tris-Cl buffer (pH 8.5), and stored at −20 °C. Viral assemblages ($n = 12$) were obtained via sequential filtration of 20 L seawater per sample, followed by iron chloride flocculation[66], with modifications for prevention of DNA degradation and removal of PCR inhibitors[23]. Briefly, peristaltic pumps and 142 mm rigs were used to remove the cellular fraction via sequential filtering through glass fibre (GF/D: pore size 2.7 μm) then polyethersulfone (pore size 0.22 μm) filters, before flocculation and precipitation of viruses via iron chloride. Iron-bound viral particle flocculate was recovered onto 1.0 μm polycarbonate filters (within 4 h of collection); filters were then stored in the dark at 4 °C. Viruses were resuspended (within 5 months of collection) in ascorbate-EDTA buffer

(0.1 M EDTA, 0.2 M $MgCl_2$, 0.2 M ascorbic acid, pH 6.1), concentrated using Amicon Ultra 100 kDa centrifugal filter units (Millipore UFC910024) and purified with DNase I (to remove un-encapsulated DNA). Viral DNA was extracted using the Wizard® DNA Clean-up System (Promega A7280). In contrast to previous samples derived from a coastal station (Western English Channel site L4[23]; Sargasso Sea viral metagenomic DNA required further removal of PCR inhibitors prior to successful sequencing library preparation. This was accomplished using silica membrane spin columns (DNeasy PowerClean Pro Cleanup Kit; Qiagen 12997-50), before elution in 10 mM Tris-Cl buffer (pH 8.5), and storage at 4 °C (colder storage temperatures were avoided to prevent freeze-thaw shearing of DNA for long-read sequencing library preparation).

### Library preparation, amplification and sequencing

For short read sequencing, 1 ng of host community DNA and viral assemblage DNA was used to generate Nextera XT libraries (Illumina; manufacturer's protocol); After amplification (12 cycles) and assessment of library quantity (Qubit; ThermoFisher) and quality (Bioanalyzer; Agilent), library DNA was sequenced as 2 × 300 bp paired-end sequence reads, on a HiSeq 2500 (Illumina Inc.) in rapid mode, by the Exeter Sequencing Service (University of Exeter, UK). In addition, long-read sequences (mean average length ~4 Kbp; Supplementary Table 2) were generated from viral metagenomic DNA via nanopore sequencing (Oxford Nanopore Technology: ONT) using the VirION pipeline[23], dx.doi.org/10.17504/protocols.io.p8fdrtn). Briefly, ~100 ng of DNA per sample was sheared to ~8 kbp fragments to maximise PCR and sequencing efficiency and amplified using PCR-adapters for Linker Amplified Shotgun Library (LASL) generation. Samples from 200 m did not amplify sufficiently for preparation of long-read sequencing libraries, therefore downstream analysis focused on long-read assemblies and analysis of 80m samples. Three long read viral samples from 80m were prepared using the SQK-LSK109 kit and barcoded with native barcoding before being sequenced on a single MinION Mk 1B flowcell (FLO-MIN106; R9.4 SpotON; ONT)

## Sequence processing and assembly

The following bioinformatic pipeline is summarised in Supplementary Fig. 1. Raw metagenomic short reads from both cellular and viral fractions were initially processed with cutadapt[67] to remove adaptors and PhiX reads (i.e., control library), then error-corrected and quality controlled with bbmap (https://jgi.doe.gov/data-and-tools/bbtools/). Cleaned reads for each sample were assembled independently with metaSPAdes (v3.13.1[68] using k-mer sizes: 21, 33, 55 and 77. Long-read sequence data was basecalled with high accuracy using Guppy v3.3.0 and demultiplexed with Porechop (v0.4; https://github.com/rrwick/Porechop). Fewer than 42% of the reads were successfully assigned to a barcode, so following removal of adapters and barcodes with Porechop (including removal of reads with adapters in the middle), all reads from the three 80m samples were pooled together and filtered with NanoFilt[69] to remove those with a q-score < 10. Pooled, clean, high quality long reads were assembled (via Overlap Layout Consensus: OLC) with metaFlye[70] and Minipolish (including 10 rounds of Racon)[71,72], followed by an additional polishing round with Medaka (https://nanoporetech.github.io/medaka/) and two rounds of short-read polishing with cleaned and pooled short reads from matched samples[73].

## SAR11 and Prochlorococcus relative abundances

Metabarcoding 16S rRNA amplicon sequencing and analysis was performed as follows: Four litres of seawater were filtered onto 0.2 μm Sterivex filters. DNA was extracted using a phenol-chloroform protocol[74] and amplified with V1-V2 primers 27F (5′-AGAGTTTGAT CNTGGCTCAG-3′) and 338RPL (5′-GCWGCCWCCCGTAGGWGT-3′). Libraries were sequenced using 2 × 250 Pair-End on the MiSeq platform (Reagent Kit v2). The 16S rRNA sequences were trimmed and dereplicated, chimera checked, and ASVs generated using the DADA2 R[75] package, version 1.2[76]. Taxonomy was assigned with the Silva database version 123[77]. Amplicon datasets collected at 80 m and 200 m depth from the BIOS-SCOPE Cruise AE1712 core casts were extracted with Phyloseq[78] and relative contributions were plotted using the ggplot package[79] in R[75] (above pipeline as described by ref. 80). Amplicon raw data SRAs used in this study are available at NCBI Bioproject PRJNA769790.

## Viral sequence recovery and dereplication

To identify Sargasso Sea viruses in our assemblies, contiguous sequences from long and short read assemblies that were circular or ≥10 kb were processed with VirSorter (v1.0.5[29]) after augmenting its database with the Xfam database of viral HMM profiles from Guo et al.[81]; contigs resolved into categories 1, 2, 4, and 5 were classed as putatively viral and passed into downstream analyses[24]. One viral sample (collected July 8th from 200 m) was excluded from further investigation due to a lack of viral sequence detection, the result of very low sequencing depth (29.6 Mbp, 100 times smaller than that of other samples). Viral contigs were dereplicated into viral populations by clustering those that shared ≥95% nucleotide identity across ≥85% of the contig length, using ClusterGenomes (https://github.com/simroux/ClusterGenomes)[24]. The longest contig within a cluster was selected as the cluster (population) representative. Where populations were represented by contigs derived from long reads, the presence of fragmentation in short-read assemblies was investigated by mapping: I) population members derived from short reads; II) all short-read contigs >1 kb (Minimap2[82]). The results of mapping were parsed with a custom-written Python script to identify regions of the representatives that were not covered by short-read assemblies ('calc_breakages.py'; available from https://github.com/BIOS-SCOPE/AE1712-viromes). Using the R[75] packages ggplot[79], tidyverse[83], cowplot[84], colorspace[85] and ggrepel[86], fragmentation (number of breakages) and recovery (percentage aligned) of the representatives were plotted against their relative abundances (RPKM values in short-read data calculated with

CoverM (v0.2.0) (https://github.com/wwood/CoverM)). Cluster representatives were used in a second round of clustering with a combined dataset of Global Ocean Viromes 2.0 dataset (GOV 2.0[30]; accession numbers ENA: PRJEB402; PRJEB9742; NCBI: PRJNA366219) to identify viruses that belonged to known viral populations in Global Ocean datasets and those that were novel to the Sargasso Sea.

## Metagenome assembled genomes

To increase likelihood of host-prediction of phage contigs, the genomes of cellular Sargasso Sea microbes were assembled as follows: Short reads were mapped to the contigs from the cellular fraction using CoverM (v0.2.0) (https://github.com/wwood/CoverM). Reads were retained if they had > 95% identity and > 75% read coverage with trimmed_mean used to remove the top 5% and bottom 5% depths. The bam files from read-mapping and the contigs from cellular samples were used as inputs into a custom script for binning. First, UniteM (v0.0.18) (UniteM; unpublished https://github.com/dparks1134/UniteM) was used to create a set of initial bins using the unitem bin command, utilizing the following binning options: gm2, bs, mb2, max40, max107, mb_verysensitive, mb_sensitive, mb_specific, mb_veryspecific. From this set of bins, UniteM profile and UniteM consensus commands were used to produce an ensemble bin set. Concordantly, DAS Tool (v1.1.1)[87] and the MetaWRAP (v1.0.6)[88] bin_refinement module were utilized on the bins produced from the unitem bin command to produce ensemble bins. However, the MetaWRAP (v1.0.6) bin_refinement module only accepts three candidate bin sets, so MetaBAT2[89], GroopM2[90], and MaxBin2[91] outputs from UniteM were used as input into the MetaWRAP module. After all ensemble binning techniques were complete (MetaWRAP, DAS Tool, UniteM), the product ensemble bins were used as input for UniteM, MetaWRAP, and DAS Tool for a second iteration to produce an optimal bin set. Following the second iteration of ensemble binning, the output bins of each tool individually was evaluated for completeness and contamination using the CheckM (v1.0.12)[92] lineage workflow. Once the completeness and contamination statistics for the bin sets of the second iteration of ensemble binning tools were obtained, the bins greater than or equal to 70% completion and less than or equal to 10% contamination were used to calculate a quality score. Similar to the methods used in UniteM and CheckM, a score was calculated as the following: score = completeness − (2 x contamination). Each ensemble binning tool was scored, and then the tool with the highest quality score was used as the bin set for that particular sample. Following scoring, RefineM (v0.0.24)[93] outliers was used to remove any potential outliers associated with GC content or tetranucleotide signatures with the following parameters: --gc_perc 95 –td_perc 95. The taxonomical classification of the resulting 89 MAGs was undertaken with GTDB-tk18[94]; genomes were classified via placement in a GTDB reference tree.

## Local and global relative abundance calculations

Next, we investigated which viruses were most abundant in our samples, and how Sargasso Sea viruses are distributed across the Global Ocean. For local abundance calculations, Illumina reads that passed quality controls were competitively recruited to the Sargasso Sea dataset of viral population representatives (derived from both VirION and short-read assemblies) using Bowtie2[95], in non-deterministic, sensitive mode; the resulting bam files were parsed in BamM (https://github.comEcogenomics/BamM) to retain reads that mapped at ≥90% read length at ≥95% identity[24]. The abundance of viral populations within each sample were calculated using mean contig coverage (excluding <5th and >95th percentile – tpmean[30] using BamM coverage. Population representatives with < 70% coverage[30] within a sample were assigned an abundance of 0 to minimise false positive detection of populations within a sample[24]. Coverage values of viral populations were then normalized by total number of reads per metagenome as a proxy for relative abundance. Total relative abundance of long-read derived and short-read derived viral populations, and differences

between the lengths of those viral populations, was tested and plotted with the R[75] packages tidyverse[83], cowplot[84], and scales[96]. Local diversity of the Sargasso Sea viral populations from cellular fraction and viral fraction samples was assessed using the packages vegan[97] (https://CRAN.R-project.org/package=vegan) and pracma (https://cran.r-project.org/web/packages/pracma/index.html) in R[75]. Community structure of Sargasso Sea viruses was evaluated with Principal Coordinates Analysis (PCoA) based on Bray-Curtis dissimilarity (function vegdist) from cube-root transformed abundance of representative viral contigs. Clusters were tested for statistical significance using standard deviation and F-tests (function Adonis; 999 permutations); ellipses were visualised (function ordiellipse) at 95% (inner) and 98% (outer) confidence intervals. Presence-absence of viral populations in cellular fraction and viral fraction samples was visualised using Adobe Illustrator[98].

To investigate the global distribution of the recovered Sargasso Sea viral populations, short reads from this study and from the GOV 2.0 dataset[30] were mapped back to the Sargasso Sea and GOV2 representative viral population contigs. The GOV 2.0 dataset contains 145 samples from five distinct global ecological zones, including the Arctic, Antarctic, bathypelagic, temperate and tropical epipelagic, and mesopelagic. Datasets were subsampled to 5 million reads prior to read mapping to prevent sequencing depth influencing the likelihood of contigs meeting the minimum genome coverage cut-off value ($\geq 70\%$) and thus possible inflation of the number of rare viruses detected as present in larger datasets compared to smaller datasets. Subsampled reads were recruited against a dereplicated set of Sargasso Sea and GOV2 viral population contigs (using the cut-offs and read recruitment strategy detailed above). Estimated presence/absence values of Sargasso Sea viruses were then calculated singly for each GOV2 site, and for the dataset as a whole. Presence/absence of Sargasso Sea viral populations in the GOV2 dataset were plotted using the ggplot package[79] in R[75], as were normalised (between sample sites) levels of nitrogen (NO3 = Nitrate+Nitrite-1 (umol/kg) and Phosphate (PO4 = Phosphate-1 (umol/kg). Global distribution of Sargasso Sea samples was visualised with the R[75] packages Simple Features[99] and rnaturalearth (https://github.com/ropensci/rnaturalearth), and the vector graphics editor Inkscape[100].

## Viral classification, microbial survey and host prediction

Having examined the abundance of Sargasso Sea viruses, we next investigated which viruses were present, how the bacterial community was structured, and whether we could predict the hosts of our viral genomes. To classify viruses, open reading frames (ORFs) in viral population representatives were identified with Prodigal (v2.6.1)[101] in metagenomic mode with default settings. Population representatives were clustered into ICTV-recognized viral genera using vConTACT2[102] alongside RefSeq prokaryotic viral genomes (release 88) for reference to known isolates.

Linkage of viral populations to putative hosts was attempted as follows: Putative hosts were assigned to viral populations through prophage blast, tRNAscan-SE (v1.23)[12], and WisH (v1.0)[13] using a scoring approach similar to what has been previously reported for human gut viromes[14]. In prophage blast, a nucleotide blast database was built by using Sargasso Sea MAGs. Viral representative contigs were used as input to BLAST against this database. Scores ranging from 1 to 4 were assigned based on percent identity and coverage (4: 98% ID and 90% cov, 3: 90% ID and 75% cov, 2: 90% ID and 50% cov, 1: 90% ID and 30% cov). General tRNA models were predicted for viral contigs in tRNAscan-SE (v1.23). Secondary structures of MAGs were searched using bacterial tRNA model. Scores were assigned to hits according to percent identity (3: 100%, 2: 95%, 1: 90%). Host models were built in WisH (v1.0). Null models were predicted by using 283 decoy RefSeq viral sequences that infect non-marine isolates belonging to the genera Staphylococcus, Streptococcus, Lactobacillus, Propionibacterium,

Mannheimia, and Paenibacillus since none of these genera should encompass ocean MAGs. The virus-host linkages were predicted by providing target viral contigs, host MAG models, and the matrix of null model parameters. Scores were given based on reported p-values ($p$-value $\leq 10$-10: 2.5, $p$-value $\leq 10$-5: 2). Collectively, virus-host linkages that had scores $\geq 3$ were considered as putative hosts. As just 0.26% of viral contigs were successfully linked to hosts, alternative methods of host prediction were then attempted.

Recalling previous evidence that at Sargasso Sea, *Prochlorococcus* viruses may be more important to total viral abundance than SAR11 viruses (*sensu*[11]), we evaluated evidence of these phages in Sargasso Sea viromes using two approaches: First, we determined if assembled contigs from Sargasso viromes could be associated with SAR11 or *Prochlorococcus* hosts. The top 100 most abundant viral populations were screened for marker genes associated with cyanophages and pelagiphages using DRAM-v[103]. Specifically, contigs containing photosynthetic *psbA* gene (prevalent in cyanophages)[104] were extracted as putative cyanophages for manual curation and assessed for completeness with CheckV v0.3.0[105]. Cultured pelagiphages lack appropriate signature auxiliary metabolic genes which could putatively be used to identify pelagiphages from metagenomic data. However, terminase (*TerL*) genes are commonly used to construct pelagiphage phylogenies[13,106] which correlate to clustering of pelagiphage genomes from shared-genenetworks (e.g. vConTACT2)[106]. To identify pelagiphages, DRAM-v annotations that specified terminase (*TerL*) genes were identified from published pelagiphage genomes, non-pelagiphages (from NCBI refseq, search term: 'marine terminase in viruses'), *Pelagibacterales* (identified via BLAST hits against terminase from known pelagiphages) and Sargasso Sea viral populations. *TerL* genes were aligned using E-INS-i strategy for 1000 iterations in MAFFT v7.017[107]. Aligned sequences were trimmed with Trimal v1.4.rev15[108] with sites containing more than 50% gaps removed from the alignment. Alignments were manually checked for overhangs in Geneious v10.2.6[109]. After determining an appropriate substitution model using Model Finder[110], a phylogenetic tree was constructed using IQ-tree[111] with rapid bootstrap support generated from 1000 iterations. The phylogeny was visualized in iTOL v5[112], and each clade was subsampled to improve clarity whilst retaining diversity within the tree. Sargasso Sea populations which clustered more closely to non-pelagiphages than pelagiphages were considered to demonstrate dissimilarity to known viruses of SAR11.

Next, we assessed whether genomes similar to those of previously isolated viruses of cyanobacteria and SAR11 were represented in Sargasso Sea viromes through mapping of short reads. High quality short reads were mapped against a dereplicated set of all published cyanophage and pelagiphage isolate genomes (accession numbers: Supplementary Dataset 4). Viral population representatives were generated using ClusterGenomes[24] as above (cut-offs $\geq 95\%$ nucleotide identity across $\geq 85\%$ genome length). *Escherichia* phage T4 was added as a negative control. Reads were recruited using Bowtie2[95], in non-deterministic, sensitive mode, and the resulting bam files were parsed in CoverM (https://github.com/wwood/CoverM) to retain reads that mapped at $\geq 90\%$ read length at $\geq 95\%$ identity[24]. RPKM was calculated as a proxy for relative abundance. To evaluate whether detection of phage isolates was sensitive to minimum genome coverage cutoffs, we conducted this analysis using minimum coverage thresholds of 40% and 70%. Lastly, to compare the representation of previously isolated viruses of cyanobacteria and SAR11 in Sargasso Sea viromes to previously published Sargasso Sea samples and global viromes, we repeated this analysis using previously published metagenomes from the Sargasso Sea[56] (cellular-fraction only; no quantitative viromes of dsDNA viruses from the Sargasso Sea has been previously published), and the GOV 2.0 dataset. Data manipulation and plotting was conducted in R[75], using packages tidyverse[83], cowplot[84], and colorspace[85].

## Inter-community diversity calculations

Evaluation of the inter-community diversity of the Sargasso Sea viral populations in relation to global viral populations was conducted as follows: Filtered and sequencing-depth normalized read mappings of the complete GOV2 dataset and Illumina reads from BATS against a combined, dereplicated database of Sargasso Sea (this study) and GOV 2.0[30] viral populations ($n = 152,979$) for abundance calculations (above) were processed to evaluate viral inter-community diversity using the vegan package[97] (https://CRAN.R-project.org/package=vegan) in R[75]. Viral β-diversity and community structure was evaluated with PCoA based on Bray-Curtis dissimilarity (function vegdist) from cube-root transformed abundance of representative viral contigs. Statistical support for clusters was evaluated using standard deviation and F-tests (function Adonis; 999 permutations) and visualised with ellipses drawn at 95% (inner) and 98% (outer) confidence intervals (function ordiellipse). The PairwiseAdonis command from package pairwiseAdonis[113] in R[75] was used to generate pairwise comparisons between the groups: centroids were calculated from the mean of the samples' weighted averages within each group along the unconstrained ordination axes, and Euclidean distances were estimated for each pair of centroids. A Similarity Percentages (SIMPER) analysis (function adonis) was conducted to identify the key Sargasso Sea viral populations driving the community structure depicted via PCoA. To ascertain if the most important contributors to β-diversity were comprised of globally abundant or Sargasso-Sea specific viruses, a bootstrap ($n = 10000$) test was conducted of the median number of Global Ocean Virome (GOV2) samples in which each Sargasso Sea virus appeared (code for bootstrapping available from: [https://raw.githubusercontent.com/btemperton/tempertonlab_utils/master/R/StatsUtilities.R][114]). Global relative abundance of Sargasso Sea long-read viral contigs and isolates that infect SAR11 and Prochlorococcus were plotted using R[75] packages tidyverse[83],cowplot[84], scales[96], and ComplexHeatmap[115], with editing in Inkscape[100] for addition of Ocean and sea regions.

## Microdiversity calculations

We next evaluated microdiversity in Sargasso Sea virome short- and long-read data to investigate: 1) whether the nutrient-limited waters of the Sargasso Sea were enriched in microdiverse viral populations typically associated with hosts that favour such conditions; 2) whether microdiversity was significantly different between viral populations from 80 m and 200 m; 3) whether previous findings that long-read viromes capture greater microdiversity in the viral fraction in a coastal region were similarly observed in viromes from nutrient-depleted environments[23,26]. To compare average microdiversity (nucleotide diversity: π[60]; between in Sargasso Sea viruses and those captured in the Global Ocean Virome (GOV2), we used the same approach as Gregory et al.[30]. Long-reads (not available in GOV2 datasets) were excluded from this analysis to avoid potential artefacts associated with sequencing technology. Briefly, all short-read Sargasso Sea viromes were randomly subsampled without replacement to 1M reads using bbmap (https://sourceforge.net/projects/bbmap/). The subsampled reads were assembled, viruses identified, and relative abundances calculated (all methods as above) to generate BAM files. BAM files extracted from Bowtie2 with default parameters were used as inputs to metapop[116] to call single nucleotide variants (SNVs). Viral populations were only included if ≥70% of representative contigs were covered with an average depth of ≥10X. SNVs with a quality call of > 30 (QUAL score; phred-scaled) were retained, and only those with alternative alleles with a frequency > 1% and supported by ≥ 4 reads were regarded as SNV loci. To minimize sequencing errors and address coverage variations, coverage was randomly subsampled to 10X coverage per locus across the genome. To calculate average microdiversity in Sargasso Sea viruses for comparison to Global Ocean Virome (GOV2) and between depths, mean π from 80m and 200m samples were calculated with 1000 bootstraps of 100 randomly subsampled π values from short-read Sargasso Sea viromes (with replacement). Differences in mean π between depths in Sargasso Sea samples were compared to a null model in which π values from both depths belong to one population (by randomly shuffling of labels and splitting into two datasets of equal size to initial datasets). To investigate whether microdiversity in VirION-derived viral genomes was significantly higher than those in short-read only viral genomes, preliminary work included the generation of additional BAM files by mapping short reads to short-read and VirION assemblies (together; method as above), and the calculation of 100 permuted π values from Sargasso Sea viruses assembled using each approach (i.e., short reads only/VirION pipeline) to generate distributions within those permutations. The permuted percentage increase between the mean π values from each approach was then tested for significance via permutation and bootstrapping test (1000 iterations), as above. The R[75] packages used for data manipulation, permutation and bootstrapping tests and plotting results were Tidyverse[83], Cowplot[84], Colorblindr(https://rdocumentation.org/packages/colorblindr/versions/0.1.0), Colorspace[85], and Scales[96].

## Investigation of virus hypervariable regions

To investigate the presence and content of hypervariable regions, high quality short reads were mapped to the top 50 most abundant Sargasso Sea viral population representatives (derived from both VirION and short-read assemblies) using Bowtie2[95]. The resulting bam files were filtered to retain quality mappings (at 95% identity and 70% coverage) using CoverM (https://github.com/wwood/CoverM). The per-nucleotide coverage of the top 50 most abundant viruses was generated from the filtered bam files using bedtools2 (https://github.com/arq5x/bedtools2) and used as input to find HVRs, which were defined as regions that possessed: less than 20% of the whole contig median coverage; at least 600 bp; zones of zero coverage[117]. Functions encoded within candidate HVR regions were investigated using a tBLASTx search against the NCBI NR database.

## Reporting summary

Further information on research design is available in the Nature Portfolio Reporting Summary linked to this article.

## Data availability

The sequencing data and assemblies generated in this study have been deposited in the National Center for Biotechnology Information (NCBI) database under the BioProject accession code PRJNA767318. The amplicon raw SRA data used in this study are available in the NCBI database under the BioProject accession code PRJNA769790. The following data generated in this study are provided in the Supplementary Information: PERMANOVA analysis results (Supplementary Dataset 1); 'BATS-B' (from SIMPER analysis) contig abundance (Supplementary Dataset 2); Metagenome Assembled Genomes (MAGs) (Supplementary Dataset 3); Accession numbers of published cyanophage, pelagiphage and T4 genomes used in the production of Supplementary Fig. 7 (Supplementary Dataset 4). The following data generated in this study are provided in the Source Data file: Alignments of Sargasso Sea long-read derived viral population representatives against short-read derived population members; used as input for the script calc_breakages.py, towards production of Supplementary Figs. 2A and B (alignment_short_members_t0_long_reps.txt); Areas of Sargasso Sea long-read derived viral population representatives not aligned to short-read population members; output of calc_breakages.py, towards production of Supplementary Fig. 2A, B (LRR_breakages_2023_02_23.csv); Relative abundance (RPKM) of Sargasso Sea long-read derived viral population representatives in short-read data; towards production of Supplementary Fig. 2A, B (long_read_cluster_rep_rel_abundance_covminzero.txt); Alignments of Sargasso Sea long-read derived viral population representatives against short-read contigs <1kb in length;

used as input for the script calc_breakages.py, towards production of Supplementary Fig. 2C (alignment_shortread_contigs_gr1kb_to_all_LRR-sorted-by-target-name.txt); Areas of Sargasso Sea long-read derived viral population representatives not aligned to short-read contigs <1kb; Output of calc_breakages.py; towards production of Supplementary Fig. 2C (breakages_all_LRR_contigs_gr_1kb.csv); Relative abundance (RPKM) of Sargasso Sea long-read derived viral population representatives in short-read data; towards production of Supplementary Fig. 2C (All_long_read_cluster_rep_RPKM.txt); Abundance tables generated from mapping Sargasso Sea reads and GOV2 reads to Sargasso Sea population representative contigs, plus GOV2 metadata; used in production of Figs. 3 and 5A. (folder figure5a_figure3); Microdiversity values for Sargasso Sea contigs used to produce Supplementary Figs. 9, 10, and 11 (sl_contig_microdiversity.tsv); Viral population representative contig names, lengths and type; used for production of Fig. 1 (length_2301.txt); Rank and abundance values for short-read viral population representatives; used for production of Figs. 1, 5 ('rank_abundance_2301.csv); Metadata (including labels) for GOV2 and Sargasso Sea samples; used for production of Fig. 3 (BATS_GOV2.0_env.cvs); Short-read coverage of GOV2 and Sargasso Sea viral populations; used for production of Fig. 3 (GOV2.0_BATS_coverage); Metadata (including labels) for Sargasso Sea samples; used for production of Fig. 5 and Supplementary Figs. 3, 4 and 5 (BATS_env.csv); Short-read coverage of Sargasso Sea viral populations; used for production of Fig. 5A (BATS_short_reads_2031_coverage_rm2v.csv); Short-read coverage of Sargasso Sea viral populations; used for production of Supplementary Fig. 3A (GOV2.0_BATS_coverage_S3A.csv); Short-read coverage of Sargasso Sea viral populations; used for production of Supplementary Fig. 3B (GOV2.0_BATS_coverage_S3B.csv), Short-read coverage of Sargasso Sea viral populations from short-read sequencing; used for production of Supplementary Fig. 4 (BATS_short_reads_1044_coverage.csv) Source data are provided with this paper.

## Code availability

Code for analyses and associated data used in the current study are available at: https://github.com/BIOS-SCOPE/AE1712-viromes, https://doi.org/10.5281/zenodo.10940125[118].

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

## Acknowledgements

The authors thank the crew and the marine technicians of the Bermuda Institute of Ocean Science vessel, the Atlantic Explorer, for the collection of seawater samples. Bioinformatic analyses were conducted using the high-performance computing resources of the Ohio Supercomputer Center provided by the Louisiana State University and those of ISCA, provided by the University of Exeter. The following grant information was disclosed by the authors: Major support was provided by a fellowship to Ben Temperton from the Bermuda Institute of Ocean Sciences as part of the BIOS-SCOPE program (Simons Foundation International); the Royal Society and the Natural Environment Research Council (NERC) (NE/P008534/1 and NE/R010935/1). Additional support to Joanna Warwick-Dugdale from a NERC GW4+ Doctoral Training Partnership PhD (NE/L002434/1), and the BIOS-SCOPE program. Work conducted by Luis Bolanos and Rachel Parsons were funded by the BIOS-SCOPE program. The work performed by Holger Buchholz was funded by a NERC GW4+ Doctoral Training Partnership PhD. Funding support for Matthew B. Sullivan included Gordon and the Betty Moore Foundation (awards

#3790 and 5488) and the National Science Foundation (NSF) (OCE#1829831, ABI#1759874, and OCE#1829640). Support for this project also came from the NSF Center for Chemical Currencies of a Microbial Planet (C-CoMP NSF-STC 2019589). This is C-CoMP publication #041. This project utilised equipment funded by the Wellcome Trust Institutional Strategic Support Fund (WT097835MF), Wellcome Trust Multi-User Equipment Award (WT101650MA) and BBSRC LOLA award (BB/K003240/1). BT: BIOS-SCOPE (Established 2015) - Simons Foundation International; Royal Society; Natural Environment Research Council (NERC): NE/P008534/1; NE/R010935/1. JWD: BIOS-SCOPE (Established 2015) - Simons Foundation; International Natural Environment Research Council (NERC): NE/L002434/1. RP: BIOS-SCOPE (Established 2015) - Simons Foundation. LB: BIOS-SCOPE (Established 2015) - Simons Foundation. HB: Natural Environment Research Council (NERC) GW4+ PhD. MS: Gordon and Betty Moore Foundation: 3790; Gordon and Betty Moore Foundation: 5488; National Science Foundation (NSF): OCE#1829831; National Science Foundation (NSF): OCE#2019589; National Science Foundation (NSF): ABI#1758974; National Science Foundation (NSF): OCE#1829640. There was no additional external funding received for this study. The funders had no role in study design, data collection and analysis, decision to publish, or preparation of the manuscript. For the purposes of open access, the author has applied a Creative Commons Attribution (CC BY) licence to any Author Accepted Manuscript version arising.

## Author contributions

Joanna Warwick-Dugdale performed the experiments, analyzed the data, prepared figures and/or tables, authored or reviewed drafts of the paper, approved the final draft. Funing Tian analysed the data, prepared figures and/or tables, authored or reviewed drafts of the paper, approved the final draft. Michelle Michelsen performed the experiments. Dylan R Cronin analysed the data. Karen Moore performed the experiments, contributed reagents/materials/analysis tools. Audrey Farbos performed the experiments. Lauren Chittick performed the experiments. Ashley Bell analysed the data and prepared figures. Holger Buchholz analysed the data. Ahmed A Zayed analysed the data. Luis Bolanos-Avellaneda analysed the data and prepared figures. Rachel Parsons contributed reagents/materials/analysis tools, authored or reviewed drafts of the paper, approved the final draft. Michael J Allen authored or reviewed drafts of the paper, approved the final draft. Matthew B Sullivan contributed reagents/materials/analysis tools, authored or reviewed drafts of the paper, approved the final draft. Ben Temperton conceived and designed the experiments, analyzed the data, contributed reagents/materials/analysis tools, prepared figures and/or tables, authored or reviewed drafts of the paper, approved the final draft.

## Competing interests

The authors declare no competing interests.
