## [Peer Review File · Nature Communications]

Long-read powered viral metagenomics in the oligotrophic Sargasso SeaReviewer #1 (Remarks to the Author):

The manuscript "Long-read powered viral metagenomics in the Oligotrophic Sargasso Sea" describes a viral diversity dataset generated from waters collected in the Sargasso Sea by combining short-read and long-read sequencing. While this approach has been previously applied by the same research group in another marine location (i.e., the Western English Channel, Warwick-Dugdale et al. 2019), the novelty in the diversity of viral populations found is remarkable (ca. 50% of viral populations lacked representation in the GOV2.0). Some viral populations seemed to be widely distributed and of potential ecological relevance. Thus, this represents a highly valuable dataset which will be likely revisited in future works, possibly when more information about potential hosts arises. The manuscript also shows some technical results regarding differences in viral populations retrieved by both sequencing technologies and between the cellular and viral fractions, which are interesting though to some point expected based on previous works (see e.g. lines 578-585), and the microdiversity of viral populations analysed.

In my view, the most critical limitation in this study is the fact that all results are based on a few of samples collected at single location during a very short time frame (4 consecutive days), which clearly impacts the ecological significance of the results obtained. Instead of acknowledging this limitation, the authors draw broad conclusions which result highly speculative and cannot be justified by the existing results (see e.g., lines 723-728, 787-809, 893-909). In this sense, some parts of the manuscript were too verbose and should be heavily shortened, both in the introduction (e.g. lines 57 to 100) and results & discussion, particularly given the limited results obtained (e.g. sections on the hypervariable regions of viral genomes and the microdiversity of viral populations).

One intriguing aspect of the dataset (and also a main result discussed in the manuscript) is that the viral community present at that sampling time seemed to be highly different from that of other oligotrophic samples, exhibiting very low abundance of pelagiphages and cyanophages. In this sense, several questions arise, which in my view have not been appropriately addressed:

1. Is this a particular feature of the set of samples analysed, or was the same result found in other viromes from the Sargasso sea? (such as the ones previously included in GOV). Were pelagiphages and cyanophages abundant in these other samples? This is not clear.
2. Based on the composition of MAGs from the cellular fraction (Table S3) it seems that SAR11 and Prochlorococcus were not abundant in the samples. Could the authors provide more information on the microbial community composition (based on miTAGs for example?) to give an idea of the abundance of these potential hosts? This is critical as a large part of the discussion is centered around the idea of a low lytic viral activity of pelagiphages and cyanophages, referring to other works as a support of the results presented (e.g. lines 47-49, 122-133, 706-711). However, the results presented here picture a very different scenario, where those viruses were not even abundant in the samples (maybe just because their hosts were missing?), so this is confusing.

Finally, other problems in the analysis are the extremely low recovery of potential hosts (< 0.3%), or hypervariable regions (only 6 contigs in the whole dataset, as shown in Figure 6), which also limits the significance of these results obtained.

As suggestions to improve, I would advise to use complementary tools to try to identify a higher number of potential hosts (e.g., software like PHIST or PHP, the use of CRISPRCasFinder, etc.), and as stated above, providing more information of the microbial communities present at the time of sampling. Additionally, not sure if there is enough resolution, but would it be possible to perform a more detailed analysis of this dataset studying in parallel the viral and cellular fraction (both for viruses and potential hosts) over the different sampling times? This may provide some clues about their short-term dynamics and help clarifying some of the unsolved questions in the manuscript. The quality of the figures should be also improved as in some cases they were difficult to visualize due to the size, or were not very informative (e.g., axis X from Figure 4 should highlight geographic regions for an easier interpretation). Finally, the manuscript should be more concise, avoiding overinterpretation of the results.

A few other specific points are mentioned below:

Line 372: "to investigate [...] whether previous findings that long-read viromes capture greater microdiversity in the viral fraction in a coastal region were similarly observed in viromes from

nutrient-depleted environments". However, the authors did not use long-read viromes to calculate microdiversity, right?

Line 429-431: Please, take into account that viral reads in the cellular fraction are not only lysogenic or attached to the cells, they could also originate from actively replicating viruses. This should be rewritten.

Line 629. "High rates of lysogeny in taxa such as SAR11 would produce a similar outcome, although the most abundant pelagiphages in the oceans do not encode known mechanisms of chromosomal integration" Please, revise this sentence after the study of Morris et al. "Lysogenic host-virus interactions in SAR11 marine bacteria"(2020, Nature Microbiology)

Reviewer #2 (Remarks to the Author):

Review for Warwick-Dugdale et al 2023

Reviewer: Cynthia Silveira

Summary

This manuscript investigates the viral community diversity in an important oligotrophic ocean, the Sargasso Sea. The Sargasso Sea has been the site of significant discoveries in the field of microbial ecology that informed much of the early development of the field, especially when it comes to Bacteria and Archaea. Yet, the viral communities in this environment remain overlooked. Therefore, this reviewer considers this work as a significant contribution to the field. Importantly, the authors introduce an approach that combines long and short read sequencing, which is rarely applied to any ecosystem. This approach led to the main findings (and strengths) of this work: long-read sequencing enabled the discovery of over 1000 viral populations that would have been missed by short reads. It also allowed the detection of microdiversity patterns and hypervariable regions in viral genomes that may be important for ecological adaptation and impact virus-host coevolution (and therefore, host community structure and function). The methods are sound and the manuscript is well-written. The main weakness is the small sample size (2 metagenomes and six viromes from two depths), but I don't think this undermines the relevance of the work. Below are specific questions and concerns, mostly for clarification.

Major comments:

An important question about the methods is whether shorter (<10Kb) contigs from the short read assemblies were aligned to long read contigs in the fragmentation analyses (Lines 472-473, and 515) or if only >10Kb contigs were used. You would expect that fragmentation would cause most contigs to be shorter than 10Kb, which would explain the large number of singleton clusters represented by long read contigs. The text mentions the recovery of 115 viral populations by these long read contigs, a small fraction of the dataset. Testing if other populations recruit shorter contigs (>1Kb) would make this result stronger and further rule out any possibility of bias from sequencing technology (which short read mapping would not eliminate).

The reads are publicly available in the NCBI, but I could not access the codes on protocols.io, they seem to be still private.

My main concern with the discussion is the extrapolation of the findings as a potential model for oligotrophic oceans in future climate scenarios. I think the small sample size, the restricted geographic and temporal scope, and the lack of associated metadata on heat penetration and mixed layers make this discussion too speculative. I think understanding viral genomics in this region of the ocean is important by itself. I suggest removing the motivation about the Sargasso Sea being a model for oceans in the Anthropocene from the abstract (Lines 28-29) and from the

discussion (Lines 781-809).

Another aspect of the discussion that is highly speculative is whether viruses of Cyano and SAR11 are not active. While I enjoyed reading the discussion of potential explanations, the fact that hosts to only 4% of viruses could be identified prevents making well-informed inferences about the killing of specific host groups. Therefore, I suggest removing that from the abstract (Lines 46-49), and keeping it in the discussion. In the abstract, instead, I suggest elaborating a bit more on the microdiversity and hypervariable regions.

Minor comments:

A minor criticism is the use of the term "ecologically important" (Lines 43, 499, 523) to describe abundant viruses. This statement contradicts the finding that most abundant viruses in the free virion fraction were not found in the cellular fraction, which may indicate they are not actively infecting.

Line 494: Was this statistically tested? If so, please report test and results.

Line 502: Report statistical tests used and results of the test. It is hard to tell the ellipses apart in the figure.

Lines 534-535: Was this difference statistically tested? From the figure, it looks like MES is actually closer.

Lines 569-570: The relative abundances of viral populations in the cellular and viral fractions should not be compared in the same PCA. The higher contribution of host DNA will make the relative abundances in the cellular fraction intrinsically smaller, and artificially separate them from the viral fraction. Figure 5b (which does not consider relative abundances, only presence-absence) tells a much cleaner story.

Line 616: The two explanations are not necessarily alternative. Low infection frequency (explanation II) may contribute to long turnover times (explanation I) if viral particles are stable in the environment.

Line 697: The coverage cutoffs seem to disagree between the main text, the figure, and the figure legend. Please, check. Also, the labels in Figure S7 are too small, I could not read.

Line 729: I believe this refers to Figure S8, not S3?

Lines 771-780 repeat Lines 635-643.

REVIEWER COMMENTS: Responses to reviewers' comments

Reviewer #1 (Remarks to the Author):

The manuscript “Long-read powered viral metagenomics in the Oligotrophic Sargasso Sea” describes a viral diversity dataset generated from waters collected in the Sargasso Sea by combining short-read and long-read sequencing. While this approach has been previously applied by the same research group in another marine location (i.e., the Western English Channel, Warwick-Dugdale et al. 2019), the novelty in the diversity of viral populations found is remarkable (ca. 50% of viral populations lacked representation in the GOV2.0). Some viral populations seemed to be widely distributed and of potential ecological relevance. Thus, this represents a highly valuable dataset which will be likely revisited in future works, possibly when more information about potential hosts arises. The manuscript also shows some technical results regarding differences in viral populations retrieved by both sequencing technologies and between the cellular and viral fractions, which are interesting though to some point expected based on previous works (see e.g. lines 578-585), and the microdiversity of viral populations analysed.

In my view, the most critical limitation in this study is the fact that all results are based on a few of samples collected at single location during a very short time frame (4 consecutive days), which clearly impacts the ecological significance of the results obtained. Instead of acknowledging this limitation, the authors draw broad conclusions which result highly speculative and cannot be justified by the existing results (see e.g., lines 723-728, 787-809, 893-909). In this sense, some parts of the manuscript were too verbose and should be heavily shortened, both in the introduction (e.g. lines 57 to 100) and results & discussion, particularly given the limited results obtained (e.g. sections on the hypervariable regions of viral genomes and the microdiversity of viral populations).

We would like to thank the reviewer for their feedback, and we accept their criticisms relating to our sampling regime limiting the ecological significance of our results. Therefore, the Introduction and Results sections have been shortened and edited as follows:

The first two paragraphs of the introduction (previously lines 57-99) have been removed, the second forming the background information for the new ‘Supplementary Discussion’ section, which contains the relocated ‘Hypervariable regions of Sargasso Sea Viral Genomes’ section (previously lines 811-890). These results are now mentioned in the main manuscript as a single sentence (lines 158-161) in the section entitled “Inclusion of long reads improves virus population recovery”.

We have modified the section entitled “Low abundance of known cyanophages and pelagiphages” to specify that the results are relevant to the sampling site and duration of the sampling regime (lines: 393-394; 405-406) and have added two sentences explaining that known cyanophages and pelagiphages may have been missed due to sampling limitations (lines 417-424).

We confirm that the high microdiversity of the viral populations obtained here is comparable to those previously reported at similar latitudes (Gregory et al., 2019), and as such this result requires discussion as to the possible mechanisms driving such observations. However, we have shortened the discussion of these results, relocating the discussion on how low host macrodiversity may promote high viral microdiversity (previously lines 787-803) to the

'Supplementary Discussion', and have removed references to our finding having implications to global viral diversity (previously lines 804-809). In addition, we have removed the final paragraph of the discussion on the significance of our results to global understanding of carbon cycling (previously 'Conclusion'; lines 893-909).

One intriguing aspect of the dataset (and also a main result discussed in the manuscript) is that the viral community present at that sampling time seemed to be highly different from that of other oligotrophic samples, exhibiting very low abundance of pelagiphages and cyanophages. In this sense, several questions arise, which in my view have not been appropriately addressed:

1. Is this a particular feature of the set of samples analysed, or was the same result found in other viromes from the Sargasso sea? (such as the ones previously included in GOV). Were pelagiphages and cyanophages abundant in these other samples? This is not clear.

Unfortunately, no other quantitative viromes of dsDNA viruses from the Sargasso Sea has been published; (Angly et al., 2006) included a Sargasso Sea virome, but used rolling circle amplification which skewed abundance estimates towards ssDNA viruses). However, we have extended our analyses to examine the evidence for cyanophages and pelagiphages in short-read viromes from the GOV 2.0 dataset, and to previously published cellular-fraction metagenomes from the Sargasso Sea. Recruitment of short reads to genomes of known cyanophage and pelagiphage isolates with minimum coverage cut-offs of 40% and 70% (new Supplementary Figure S8) revealed that known pelagiphages and cyanophages were well represented in global epipelagic viromes (GOV 2.0; TT-EPI), and less evident in mesopelagic samples (GOV 2.0; TT_MES). Cyanophages were better represented in previously published cellular-fraction metagenomes from the Sargasso Sea than in our samples, suggesting the importance of sampling timing and duration within seasonal cycles. We have added various comments to our manuscript describing these results (lines 360-366; 417-424), and the impact of sampling limitations on our results as listed above.

2. Based on the composition of MAGs from the cellular fraction (Table S3) it seems that SAR11 and Prochlorococcus were not abundant in the samples. Could the authors provide more information on the microbial community composition (based on miTAGs for example?) to give an idea of the abundance of these potential hosts? This is critical as a large part of the discussion is centered around the idea of a low lytic viral activity of pelagiphages and cyanophages, referring to other works as a support of the results presented (e.g. lines 47-49, 122-133, 706-711). However, the results presented here picture a very different scenario, where those viruses were not even abundant in the samples (maybe just because their hosts were missing?), so this is confusing.

Finally, other problems in the analysis are the extremely low recovery of potential hosts (< 0.3%), or hypervariable regions (only 6 contigs in the whole dataset, as shown in Figure 6), which also limits the significance of these results obtained.

As suggestions to improve, I would advise to use complementary tools to try to identify a higher number of potential hosts (e.g., software like PHIST or PHP, the use of CRISPRCasFinder, etc.), and as stated above, providing more information of the microbial communities present at the time of sampling.

The reviewer rightly points out that few hosts were identified in this paper for viral contigs, despite our best efforts using state-of-the art tools. Unfortunately, it is well known that accurate prediction of hosts from metagenomic-derived contigs at a taxonomic resolution

relevant to ecological function is a major challenge for the field. For instance, in a paper describing a recent tool (RaFAH), showed almost all tools perform poorly in precision and/or recall below class level (Coutinho et al., 2021). Marine metagenomes are particularly challenging due to the lack of good training data (VirHostMatcher for instance, predicts that SAR11 viruses infect *Clostridium*), a near-absence of CRISPR-Cas signature in such as SAR11 (Cameron et al., 2014), and poor correlation between virally-encoded tRNAs and associated hosts (Buchholz et al., 2022). Therefore, it is highly unlikely that use of additional tools will significantly improve the number of hosts identified to a degree that would impact the findings of the manuscript.

However, to provide a better understanding of the abundance of potential pelagiphage and cyanophage hosts during the sampling campaign, we have added a new Supplementary Figure (S7). This figure shows the relative abundances of SAR11 and *Prochlorococcus* during the sampling campaign, via the percentage of total ASVs assigned to these taxa after 16S rRNA amplicon sequencing and analysis (discussed in the manuscript on lines 341-347). We have also discussed the relevance of the host abundances reported in the Supplementary Figure S7 to the low abundance of viruses (lines 376-380).

Additionally, not sure if there is enough resolution, but would it be possible to perform a more detailed analysis of this dataset studying in parallel the viral and cellular fraction (both for viruses and potential hosts) over the different sampling times? This may provide some clues about their short-term dynamics and help clarifying some of the unsolved questions in the manuscript.

We would like to thank the reviewer for this idea, which has a lot of merit. Unfortunately, there is not enough resolution (i.e., replication; sequencing depth per sample) to statistically test for changes in the viral community of viral-fraction and cellular-fraction samples over time in this dataset. However, there is work currently underway at BATS to resolve short-term dynamics of viral communities, and we hope that future reports may shed light on the remaining questions posed by this manuscript.

The quality of the figures should be also improved as in some cases they were difficult to visualize due to the size, or were not very informative (e.g., axis X from Figure 4 should highlight geographic regions for an easier interpretation). Finally, the manuscript should be more concise, avoiding overinterpretation of the results.

In this resubmission, the figures have been uploaded in their original size and format, which should resolve visualisation issues. Figure 4 has been edited to highlight ocean and sea regions. The manuscript has been shortened including removal of overinterpretations as detailed above.

A few other specific points are mentioned below:

Line 372: “to investigate [...] whether previous findings that long-read viromes capture greater microdiversity in the viral fraction in a coastal region were similarly observed in viromes from nutrient-depleted environments”. However, the authors did not use long-read viromes to calculate microdiversity, right?

In both references supplied, the authors calculated microdiversity of long-read derived viral populations using low error-rate short reads that map to the long-read derived contigs.

Line 429-431: Please, take into account that viral reads in the cellular fraction are not only lysogenic or attached to the cells, they could also originate from actively replicating viruses.

Addition of ‘actively replicating viruses’ to line 104 (previously line 430)

Line 629. "High rates of lysogeny in taxa such as SAR11 would produce a similar outcome, although the most abundant pelagiphages in the oceans do not encode known mechanisms of chromosomal integration" Please, revise this sentence after the study of Morris et al. "Lysogenic host-virus interactions in SAR11 marine bacteria"(2020, Nature Microbiology)

We would like to thank the reviewer for pointing out this omission and have added a sentence to describing the results of (Morris et al., 2020) (lines 209-303).

Reviewer #2 (Remarks to the Author):

Review for Warwick-Dugdale et al 2023

Reviewer: Cynthia Silveira

Summary

This manuscript investigates the viral community diversity in an important oligotrophic ocean, the Sargasso Sea. The Sargasso Sea has been the site of significant discoveries in the field of microbial ecology that informed much of the early development of the field, especially when it comes to Bacteria and Archaea. Yet, the viral communities in this environment remain overlooked. Therefore, this reviewer considers this work as a significant contribution to the field. Importantly, the authors introduce an approach that combines long and short read sequencing, which is rarely applied to any ecosystem. This approach led to the main findings (and strengths) of this work: long-read sequencing enabled the discovery of over 1000 viral populations that would have been missed by short reads. It also allowed the detection of microdiversity patterns and hypervariable regions in viral genomes that may be important for ecological adaptation and impact virus-host coevolution (and therefore, host community structure and function). The methods are sound and the manuscript is well-written. The main weakness is the small sample size (2 metagenomes and six viromes from two depths), but I don't think this undermines the relevance of the work. Below are specific questions and concerns, mostly for clarification.

Major comments:

An important question about the methods is whether shorter (<10Kb) contigs from the short read assemblies were aligned to long read contigs in the fragmentation analyses (Lines 472-473, and 515) or if only >10Kb contigs were used. You would expect that fragmentation would cause most contigs to be shorter than 10Kb, which would explain the large number of singleton clusters represented by long read contigs. The text mentions the recovery of 115 viral populations by these long read contigs, a small fraction of the dataset. Testing if other

populations recruit shorter contigs (>1Kb) would make this result stronger and further rule out any possibility of bias from sequencing technology (which short read mapping would not eliminate).

We would like to thank the reviewer for posing this question. The reviewer is correct: in the fragmentation analyses reviewed (Supplementary Figure S2), the short-read population members that were aligned to the long-read population representatives were >10kb, as a 10kb cut-off was employed at the VirSorter stage of the pipeline for conservatism. As per the reviewer's suggestion, we have conducted an additional analysis where all short-read contigs >1kb were mapped to all long-read viral population representatives (n = 1410), resulting in alignments for 1387 long-read population representatives. The percentage of long-read population representatives that were recovered by short-read contigs >1kb is shown against their relative abundance in the short-read data in new Supplementary Figure S2.C. The percentage of long-read population representatives that were recovered by short-read contigs >1kb was not observed to be a function of sequencing depth.

The reads are publicly available in the NCBI, but I could not access the codes on protocols.io, they seem to be still private.

The code generated in this study are now available at:

<https://github.com/BIOS-SCOPE/AE1712-viromes>

My main concern with the discussion is the extrapolation of the findings as a potential model for oligotrophic oceans in future climate scenarios. I think the small sample size, the restricted geographic and temporal scope, and the lack of associated metadata on heat penetration and mixed layers make this discussion too speculative. I think understanding viral genomics in this region of the ocean is important by itself. I suggest removing the motivation about the Sargasso Sea being a model for oceans in the Anthropocene from the abstract (Lines 28-29) and from the discussion (Lines 781-809).

We would like to thank the reviewer for their feedback, and accept the criticisms relating to how the sampling regime limits the applicability of our results to models of the Sargasso Sea in the Anthropocene, and have therefore shortened the abstract (removing lines previously numbered 28-29), and the Results, relocating the section on how low host macrodiversity diversity may promote high viral microdiversity (previously lines 787-803) to the 'Supplementary Results and Discussion', and removing all references to our findings having implications to global viral diversity (previously lines 804-809).

Another aspect of the discussion that is highly speculative is whether viruses of Cyano and SAR11 are not active. While I enjoyed reading the discussion of potential explanations, the fact that hosts to only 4% of viruses could be identified prevents making well-informed inferences about the killing of specific host groups. Therefore, I suggest removing that from the abstract (Lines 46-49), and keeping it in the discussion. In the abstract, instead, I suggest elaborating a bit more on the microdiversity and hypervariable regions.

Reference to whether cyanophages and/or pelagiphages are active in the Sargasso Sea have been removed from the abstract as suggested.

Minor comments:

A minor criticism is the use of the term "ecologically important" (Lines 43, 499, 523) to describe abundant viruses. This statement contradicts the finding that most abundant

viruses in the free virion fraction were not found in the cellular fraction, which may indicate they are not actively infecting.

Term 'ecologically important': removed from line 43 (as part of abstract restructuring); exchanged for 'prevalence' in line 182 (previously line 499); exchanged for 'ubiquitous' in line 217 (previously line 523).

Line 494: Was this statistically tested? If so, please report test and results.

The observation that Sargasso Sea viral populations tended to appear more frequently in samples from similarly warm oligotrophic was tested via logistic regression, which is now described in lines 170-175).

Line 502: Report statistical teste used and results of the test. It is hard to tell the ellipses apart in the figure.

The following added to the manuscript main text where the figure is discussed: "analysis of variance: Adonis F-test; p -value = 0.001; Figure 3 (lines 184; 217).

Lines 534-535: Was this difference statistically tested? From the figure, it looks like MES is actually closer.

We would like to thank the reviewer for catching this error. Having conducted PERMANOVA (Pairwise Adonis) comparisons of the Euclidean distance between centroids of Sargasso Sea viruses and Global Ocean Virome (GOV2) sample groups we can confirm that Sargasso Sea viruses are more similar to TT-MES samples. The manuscript has modified to include these results (lines 192-196; 381-386; Supplemental Table S2).

Lines 569-570: The relative abundances of viral populations in the cellular and viral fractions should not be compared in the same PCA. The higher contribution of host DNA will make the relative abundances in the cellular intrinsically smaller, and artificially separate them from the viral fraction. Figure 5b (which does not consider relative abundances, only presence-absence) tells a much cleaner story.

The reviewer is correct to identify that the cellular fraction is made up largely of host DNA, and that the viral fraction therefore contributed more viral DNA to the calculations of inter-community diversity. However, the strong normalisation applied here (i.e., cube-root transformation of representative viral contig abundance; lines 678-881) minimises heteroskedasticity associated with count data over a large dynamic range, so that patterns of read recruitment can be compared robustly.

Line 616: The two explanations are not necessarily alternative. Low infection frequency (explanation II) may contribute to long turnover times (explanation I) if viral particles are stable in the environment.

We agree with this observation and have altered the line 287 (previously line 616) to include low infection frequency as a possible contributing factor to long viral particle turnover times.

Line 697: The coverage cutoffs seem to disagree between the main text, the figure, and the figure legend. Please, check. Also, the labels in Figure S7 are too small, I could not read.

In the previously submitted manuscript and supporting materials, Line 697 was a sentence pertaining to Figure S9 (the phylogenetic tree of *TerI* genes). In lines 691 and 705, we referred to Figure S8 (the abundance of known cyanophages and pelagiphages in the short-read data) and discussed specific results at >15% and >65% minimum genome coverage cut-offs values. Figure S8 was an animation (.gif) that cycled through abundance values at 0-80% minimum

genome coverage cut-off values to show how known cyanophages and pelagiphages fail away with increasing minimum genome coverage cut-off requirements. To clarify these results, we have replaced the Figure S8 animation with a series of panels at 40% and 70% minimum genomes coverages. We have included GOV2 viromes for comparison to our Sargasso Sea viral fraction samples (Supplementary Figure S8.A and B), and previously published Sargasso Sea cellular metagenomes (Biller et al., 2018) for comparison to our Sargasso Sea cellular fraction samples. We have also separated cyanophage and pelagiphage results, so that label sizes are also increased.

Line 729: I believe this refers to Figure S8, not S3?

This has been corrected.

Lines 771-780 repeat Lines 635-643.

Repetition of lines has been corrected by removal (of previously numbered lines 635-643; section on decoupling now finishes at line 318).

References

- Angly, F. E., Felts, B., Breitbart, M., Salamon, P., Edwards, R. A., Carlson, C., Chan, A. M., Haynes, M., Kelley, S., Liu, H., Mahaffy, J. M., Mueller, J. E., Nulton, J., Olson, R., Parsons, R., Rayhawk, S., Suttle, C. A., & Rohwer, F. (2006). The marine viromes of four oceanic regions. *PLoS Biology*, 4(11), 2121–2131. <https://doi.org/10.1371/journal.pbio.0040368>
- Biller, S. J., Berube, P. M., Dooley, K., Williams, M., Satinsky, B. M., Hackl, T., Hogle, S. L., Coe, A., Bergauer, K., Bouman, H. A., Browning, T. J., De Corte, D., Hassler, C., Hulston, D., Jacquot, J. E., Maas, E. W., Reinthaler, T., Sintes, E., Yokokawa, T., & Chisholm, S. W. (2018). Data descriptor: Marine microbial metagenomes sampled across space and time. *Scientific Data*, 5. <https://doi.org/10.1038/sdata.2018.176>
- Buchholz, H. H., Bolaños, L. M., Bell, A. G., Michelsen, M. L., Allen, M. J., & Temperton, B. (2022). A Novel and Ubiquitous Marine Methylophage Provides Insights into Viral-Host Coevolution and Possible Host-Range Expansion in Streamlined Marine Heterotrophic Bacteria. *Applied and Environmental Microbiology*, 88(7). <https://doi.org/10.1128/aem.00255-22>
- Cameron, T. J., Temperton, B., Swan, B. K., Landry, Z. C., Woyke, T., Delong, E. F., Stepanauskas, R., & Giovannoni, S. J. (2014). Single-cell enabled comparative genomics of a deep ocean SAR11 bathytype. *ISME Journal*, 8(7), 1440–1451. <https://doi.org/10.1038/ismej.2013.243>
- Coutinho, F. H., Zaragoza-Solas, A., López-Pérez, M., Barylski, J., Zielezinski, A., Dutilh, B. E., Edwards, R., & Rodriguez-Valera, F. (2021). RaFAH: Host prediction for viruses of Bacteria and Archaea based on protein content. *Patterns*, 2(7). <https://doi.org/10.1016/j.patter.2021.100274>
- Gregory, A. C., Zayed, A. A., Conceição-Neto, N., Temperton, B., Bolduc, B., Alberti, A., Ardyna, M., Arkhipova, K., Carmichael, M., Cruaud, C., Dimier, C., Domínguez-Huerta, G., Ferland, J., Kandels, S., Liu, Y., Marec, C., Pesant, S., Picheral, M., Pisarev, S., ... Roux, S. (2019). Marine DNA Viral Macro- and Microdiversity from Pole to Pole. *Cell*, 177(5), 1109-1123.e14. <https://doi.org/10.1016/j.cell.2019.03.040>

Morris, R. M., Cain, K. R., Hvorecny, K. L., & Kollman, J. M. (2020). Lysogenic host–virus interactions in SAR11 marine bacteria. *Nature Microbiology*, 5(8), 1011–1015.
<https://doi.org/10.1038/s41564-020-0725-x>

Reviewer #1 (Remarks to the Author):

The authors have made a thorough revision of the manuscript by significantly improving the main text, which is now more focused, and avoiding some of the problems identified in the first version (e.g., overstatements). I also appreciate that the authors have included metabarcoding data in this new version to show that the reported low abundance of known cyanophages and pelagiphages was not explained by a low abundance of their hosts in these samples. Unfortunately, despite the improvements done in the manuscript, I believe that my main concerns in terms of the novelty and ecological significance of the results still remain.

A main result/conclusion of this work is that Sargasso viral communities have a distinct community structure as compared to that of other regions. However, the very limited resolution of the sampling performed (a single "snapshot" over 3 days-sampling) remains a strong limitation. As the authors acknowledge, cyanophages were better represented in previously published metagenomes from the Sargasso sea (lines 417-424), and therefore this cannot be taken as a general rule. The fact that the composition of the phage community found in this sampling is more similar to that of mesopelagic samples is also intriguing. In the absence of any potential explanation (mixing with deep waters, or other physical events), the reason why the authors found such a different viral composition is unclear. However, I do not think it can be regarded as an evidence of a (global) differentiation of Sargasso sea viral communities from that of other regions, which would require a more comprehensive analysis.

A second major conclusion is the idea of low viral contribution to cellular turnover of SAR11 and cyanobacteria in Sargasso Sea samples. Given the problems to identify potential hosts, how can the authors discard the possibility that some of the widely distributed viruses found in this study are indeed cyanophages or pelagiphages, genetically different from the isolated ones?. Also, why do the authors restrict their comparison of the abundance of new phage populations with isolated viral genomes (e.g., HTVC010P, line 353, Figure 4). We now know that HTVC010P is actually much less abundant than other pelagiphages such as vSAG-37-F6, so the idea of HTVC010P being the most abundant virus one Earth is quite outdated. Even if the authors raise some interesting hypotheses, suggesting e.g., that highly prevalent viruses could target copiotrophs, this is still very speculative.

In general, as I mentioned in my previous review, I believe that this work is technically sound and provides an important dataset by untapping a large number of novel phage populations (including some highly abundant), which will be very useful for future studies. However, the technical approach used is not novel (see Warwick-Dugdale et al. 2019), and in my view, the conclusions drawn are still very limited in terms of ecological significance. Making progress in this field will require increasing ecological or mechanistical information in parallel with the improvement of sequencing datasets.

Reviewer #2 (Remarks to the Author):

Review for Warwick-Dugdale et al 2024

The authors satisfactorily addressed all of my concerns. Specifically, they tested the effect of 1Kb contig recruitment (new Supplementary Figure S2.C), checked statistical results (and corrected when necessary, Table S2), made their code available, and significantly revised the discussion so as not to overextend their conclusions.

REVIEWERS' COMMENTS Responses to reviewers' comments

Reviewer #1 (Remarks to the Author):

The authors have made a thorough revision of the manuscript by significantly improving the main text, which is now more focused, and avoiding some of the problems identified in the first version (e.g., overstatements). I also appreciate that the authors have included metabarcoding data in this new version to show that the reported low abundance of known cyanophages and pelagiphages was not explained by a low abundance of their hosts in these samples. Unfortunately, despite the improvements done in the manuscript, I believe that my main concerns in terms of the novelty and ecological significance of the results still remain.

A main result/conclusion of this work is that Sargasso viral communities have a distinct community structure as compared to that of other regions. However, the very limited resolution of the sampling performed (a single “snapshot” over 3 days-sampling) remains a strong limitation. As the authors acknowledge, cyanophages were better represented in previously published metagenomes from the Sargasso sea (lines 417-424), and therefore this cannot be taken as a general rule. The fact that the composition of the phage community found in this sampling is more similar to that of mesopelagic samples is also intriguing. In the absence of any potential explanation (mixing with deep waters, or other physical events), the reason why the authors found such a different viral composition is unclear. However, I do not think it can be regarded as an evidence of a (global) differentiation of Sargasso sea viral communities from that of other regions, which would require a more comprehensive analysis.

We would like to thank the reviewer for their feedback, and acknowledge that the resolution of our sampling campaign necessarily confers some limitations in terms of the wider ecological significance of our results. We have further toned down some statements accordingly (lines: 28; 157; 219-220; 303). However, our previous investigation of coastal Western English Channel viruses revealed that viral contigs obtained from a single sample (via the hybrid, short- and long-read sequencing approach) were globally abundant¹. Therefore, the observation that 54.3% of the Sargasso Sea viral populations obtained here - from two depths and over three days of sampling - were not represented in global ocean datasets, remains pertinent. This reasoning has now been added to the manuscript (lines: 174-177).

We agree with the reviewer that the similarity of the Sargasso Sea viral communities recovered from 80m and 200m to global mesopelagic samples (Figure 3) is intriguing, and have postulated that the similarity in low cyanophage abundance in both cases may be involved (lines 378-383).

A second major conclusion is the idea of low viral contribution to cellular turnover of SAR11 and cyanobacteria in Sargasso Sea samples. Given the problems to identify potential hosts, how can the authors discard the possibility that some of the widely distributed viruses found in this study are indeed cyanophages or pelagiphages, genetically different from the isolated ones?. Also, why do the authors restrict their comparison of the abundance of new phage

populations with isolated viral genomes (e.g., HTVC010P, line 353, Figure 4). We now know that HTVC010P is actually much less abundant than other pelagiphages such as vSAG-37-F6, so the idea of HTVC010P being the most abundant virus on Earth is quite outdated.

The reviewer asks a salient question: we too have wondered whether some of the ubiquitous viruses from this study whose host could not be ascertained were indeed cyanophages or pelagiphages, too unlike those in culture to be recognised as such by contemporary methods. Environmental viruses may not be successfully linked to their hosts in metagenomic datasets due to limited representation in databases of isolated taxa, and although contemporary work has made encouraging inroads in culturing the viruses of SAR11², due to their relative novelty, pelagiphages are under-represented in databases such as RefSeq, compared to cyanophages which have been extensively cultured for decades^{3,4}. In order to address this conundrum we have already undertaken two separate analyses.

The majority (88%) of cyanophages identified contain the core photosystem gene *psbA*⁵, so we first sought to identify *psbA* in the top 100 most abundant viral populations. Just two contigs, the 26th and 45th most abundant, encoded this gene with a completeness of >80% (lines 365-368), thus we did not find evidence that abundant phages with unknown hosts were cyanophages. Unfortunately, there is no such signature auxiliary metabolic gene for identifying putative pelagiphages, so instead we constructed a phylogenetic tree from the terminase genes (commonly used to construct pelagiphage phylogenies^{2,6}) of known pelagiphages (from culture, and metagenomic viral contigs where host could be determined), *Pelagibacter* isolates, and our contigs. None of the *TerI* genes from the top 100 most abundant viral populations recovered here clustered with known pelagiphage genes from culture or from metagenomes (supplementary figure 8). Thus, we did not find evidence that abundant phages with unknown hosts were pelagiphages.

Abundance of Sargasso Sea cyanophages or pelagiphages could only be ascertained for the duration of the sampling regime. We recognise this limitation and have specified it in the paragraph to this effect, which, for improved transparency, has been relocated to follow the evidence presented, now coming before our speculations on the possible mechanisms behind our observations (lines: 383-389). Other statements throughout the text have also been modified to this effect (lines: 34; 82; 348; 403-404; 409).

Even if the authors raise some interesting hypotheses, suggesting e.g., that highly prevalent viruses could target copiotrophs, this is still very speculative.

The reviewer is absolutely correct: the highly speculative nature of these results are specifically stated as such in the text (line 419); the word 'possibly' has been inserted (line 416) to further drive this point home.

In general, as I mentioned in my previous review, I believe that this work is technically sound and provides an important dataset by untapping a large number of novel phage populations (including some highly abundant), which will be very useful for future studies. However, the technical approach used is not novel (see Warwick-Dugdale et al. 2019), and in my view, the conclusions drawn are still very limited in terms of ecological significance. Making progress in this field will require increasing ecological or mechanistical information in parallel with the improvement of sequencing datasets.

In common with the reviewer, we too are frustrated by the current lack of reliable information/methods for resolving the hosts and mechanisms of abundant environmental viruses, and are very much looking forward to the technical advancements in this field that will increase our understanding of viral function in the marine environment.

Reviewer #2 (Remarks to the Author):

Review for Warwick-Dugdale et al 2024

The authors satisfactorily addressed all of my concerns. Specifically, they tested the effect of 1Kb contig recruitment (new Supplementary Figure S2.C), checked statistical results (and corrected when necessary, Table S2), made their code available, and significantly revised the discussion so as not to overextend their conclusions.

We thank the reviewer for their time, and their previous thoughtful observations and suggestions for improvements. We believe our manuscript is significantly better as a result.

References

1. Warwick-Dugdale, J. *et al.* Long-read viral metagenomics enables capture of abundant and microdiverse viral populations and their niche-defining genomic islands. *PeerJ* **7**:e6800, (2019).
2. Buchholz, H. H. *et al.* Efficient dilution-to-extinction isolation of novel virus–host model systems for fastidious heterotrophic bacteria. *ISME Journal* (2021) doi:10.1038/s41396-020-00872-z.
3. Sullivan, M. B., Waterbury, J. B. & Chisholm, S. W. Cyanophages infecting the oceanic cyanobacterium *Prochlorococcus*. *Nature* **424**, 1047–1051 (2003).
4. Clokie, M. & Millard, A. Virus isolation studies suggest short-term variations in abundance in natural cyanophage populations of the Indian Ocean. *JMBA-Journal of the marine biology association* **86**, 499–505 (2006).
5. Sullivan, M. B. *et al.* Prevalence and evolution of core photosystem II genes in marine cyanobacterial viruses and their hosts. *PLoS Biol* **4**, 1344–1357 (2006).
6. Zhang, Z. *et al.* Culturing novel and abundant pelagiphages in the ocean. *Environ Microbiol* **00**, 1–17 (2020).